# WelQrate: Defining the Gold Standard in Small Molecule Drug Discovery Benchmarking

**Yunchao (Lance) Liu**[*,1], **Ha Dong**[*,2], **Xin Wang**[*,1], **Rocco Moretti**[3],
**Yu Wang**[4], **Zhaoqian Su**[5], **Jiawei Gu**[6], **Bobby Bodenheimer**[1,7,8],
**Charles David Weaver**[3,9], **Jens Meiler**[3,10,11,12,13,14], **Tyler Derr**[1,5]

## Abstract

While deep learning has revolutionized computer-aided drug discovery, the AI community has predominantly focused on model innovation and placed less emphasis on establishing best benchmarking practices. We posit that without a sound model evaluation framework, the AI community's efforts cannot reach their full potential, thereby slowing the progress and transfer of innovation into real-world drug discovery. Thus, in this paper, we seek to establish a new gold standard for small molecule drug discovery benchmarking, *WelQrate*. Specifically, our contributions are threefold: ***WelQrate* dataset collection** - we introduce a meticulously curated collection of 9 datasets spanning 5 therapeutic target classes. Our hierarchical curation pipelines, designed by drug discovery experts, go beyond the primary high-throughput screen by leveraging additional confirmatory and counter screens along with rigorous domain-driven preprocessing, such as Pan-Assay Interference Compounds (PAINS) filtering, to ensure the high-quality data in the datasets; ***WelQrate* Evaluation Framework** - we propose a standardized model evaluation framework considering high-quality datasets, featurization, 3D conformation generation, evaluation metrics, and data splits, which provides a reliable benchmarking for drug discovery experts conducting real-world virtual screening; **Benchmarking** - we evaluate model performance through various research questions using the *WelQrate* dataset collection, exploring the effects of different models, dataset quality, featurization methods, and data splitting strategies on the results. In summary, we recommend adopting our proposed *WelQrate* as the gold standard in small molecule drug discovery benchmarking. The *WelQrate* dataset collection, along with the curation codes, and experimental scripts are all publicly available at WelQrate.org.

## 1 Introduction

Deep learning has revolutionized the field of drug discovery, providing advanced computational tools to predict the activity of small molecules against therapeutic targets. However, the focus of the AI community has primarily been on developing novel models, often putting less emphasis on establishing robust and standardized benchmarking practices. Ultimately, this disparity can impede the practical application of AI innovations in drug discovery [1].

---

[*]Equal contribution. Correspondence to `yunchao.liu@vanderbilt.edu`. [1]Computer Science Dept., Vanderbilt University (VU), [2]Neural Science Dept., Amherst College, [3]Chemistry Dept., VU, [4]Computer Science Dept., University of Oregon, [5]Data Science Institute, VU, [6]MD Anderson Cancer Center, [7]Electrical and Computer Engineering Dept,, VU, [8]Psychology Dept., VU, [9]Institute of Chemical Biology, VU, [10]Center for Structural Biology, VU, [11]Pharmacology Dept., VU, [12]Institute for Drug Discovery, Leipzig University (LU), [13]Computer Science Dept., LU, [14]Chemistry Dept., LU.

Typically, High-Throughput Screening (HTS) methods are prevalent for identifying promising compounds, but they are costly, time-consuming, and limited in their ability to explore the chemical space [2, 3]. Thus, computer-aided drug discovery seeks to train models on HTS data to offer a more efficient and scalable computational effort to predict the activity of compounds based on their structure, which is known as virtual screening. However, in spite of the importance on ensuring high-quality data for training these models, currently only a few datasets for virtual screening exist, such as MoleculeNet and Therapeutics Data Commons (TDC), but these datasets often suffer from issues like inconsistent chemical representations, undefined stereochemistry, and noisy experimental data. These flaws necessitate a more rigorous approach to dataset curation.

To address these challenges, we propose a new standard, *WelQrate*, for benchmarking small molecule drug discovery, the contributions of which are threefold as follow:

- *WelQrate* **Dataset Collection:** The *WelQrate* dataset collection are curated with stringent quality control measures including hierarchical curation, various filters and domain expert verification. The final dataset collection covers a diverse range of important theurapeutic target classes.
- *WelQrate* **Evaluation Framework:** The *WelQrate* evaluation framework incorporates critical aspects including high-quality datasets, featurization, 3D conformation generation, evaluation metrics, and data splits to provide a reliable basis for model comparison.
- **Benchmarking:** Examining model performance across several research questions using the *WelQrate* dataset collection, investigating how different models, dataset quality, featurization, and data split impact results.

Our work is driven by the need to ensure that AI models are evaluated on realistic and high-quality datasets, facilitating the translation of AI innovations into practical drug discovery solutions. The *WelQrate* dataset collection, along with detailed curation procedures and experiment scripts, is publicly available and maintained at WelQrate.org. We advocate for the adoption of our standardized evaluation practices and well-curated datasets to set a new gold standard in small molecule drug discovery, to ensure more reliable and realistic evaluations.

## 2 Related Work

The two most related efforts to establish benchmarks in small molecule drug discovery are:

**MoleculeNet** [4] is a collection of datasets for tasks essential to drug discovery and material design. However, MoleculeNet's datasets often contain errors such as invalid chemical structures, inconsistent chemical representations, undefined stereochemistry, and poorly defined endpoints [5] and further discussed in the supplement Sec. 1.1.

**Therapeutics Data Commons (TDC)  [6]** offers a wide range of datasets related to various therapeutic modalities and stages of the drug discovery process. Despite its contributions, TDC faces similar issues as MoleculeNet, including data quality concerns that affect the robustness of benchmarking outcomes, which are further discussed in the supplement Sec. 1.1.

To address the limitations of existing benchmarks, we introduce *WelQrate*, a standardized model evaluation framework considers critical aspects like dataset quality, featurization, 3D conformation generation, evaluation metrics and data split, providing a reliable benchmarking platform for real-world virtual screening. Besides, we introduce *WelQrate* dataset collection, a meticulously curated set of 9 datasets spanning 5 therapeutic target classes. Designed by drug discovery experts, *WelQrate*'s hierarchical curation pipeline includes primary, confirmatory and counter screens, along with rigorous domain-driven preprocessing such as PAINS filtering.

## 3 *WelQrate* Dataset Collection

*WelQrate* dataset collection is developed to be of high quality for AI development in three aspects: 1) realistic data setting; 2) clean and reliable data labels; and 3) standardized data formats, split schemes, featurization to facilitate a common ground for benchmarking. Details of each aspect are as follows:

First, a real-world HTS campaign not only includes a large number of compounds but also has a low hit rate, often estimated to be less than 1% [7]. To represent this reality, *WelQrate* dataset collection

contains datasets with compound numbers ranging from ~66K to ~300K and features realistically highly imbalanced activity labels to reflect the low hit rate (seen in Tab. 1). Additionally, *WelQrate* dataset collection covers a wide range of important therapeutic target classes. For example, G-protein Coupled Receptor (GPCR) is targeted by approximately 40% of marketed drugs [8].

Secondly, *WelQrate* dataset collection employs a rigorous curation pipeline to ensure high data quality [9]. The initial primary HTS has a high false positive rate. Therefore, in a real-world HTS campaign, a series of follow-up screens are carried out to ensure the correctness and relevance of the data. PubChem [10], a publicly accessible database of chemical molecules and their activities against biological assays, stores a vast number of these bioassays (experimental procedures). For our curation process, we meticulously reviewed the descriptions of the bioassays stored in PubChem to understand their relationships and experimental details. Hierarchical curation and various filters are utilized to ensure the final data is of high quality. See Sec. 3.2 for details on the curation process.

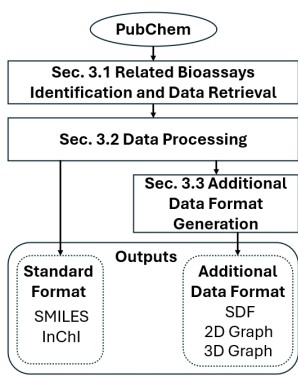

Fig. 1: An overview of the data curation pipeline.

Thirdly, *WelQrate* dataset collection provides two standard data formats and three additional data formats. Two standard formats are Simplified Molecular-Input Line-Entry System (SMILES) [11], and International Chemical Identifier (InChI) [12]. SMILES is widely used in the AI community and encodes the structure of a compound as a text sequence. Specifically, *WelQrate* dataset collection provides isomeric SMILES that contains stereochemistry information. Despite SMILES' popularity, InChI is provided as an alternative for two reasons. First, a single molecule can have multiple valid SMILES representations, which may vary between different platforms. This lack of uniqueness in representation can lead to inconsistencies. InChI, on the contrary, is generated by the InChI software to ensure uniqueness. Second, InChI can express more information than the simpler SMILES [12]. The information in InChI is organized into five layers, and we use the standard InChI, which has a prefix of "1S/".

The three additional formats are Structure Data File (SDF), 2D Graph, and 3D Graph. The 2D Graph defines edges based on bond connectivity, while the 3D Graph defines edges based on 3D Euclidean distance and includes a node attribute containing coordinate information. These additional formats are provided to facilitate fair benchmarking (See Sec. 4.1). Nevertheless, users still have the flexibility to generate their own formats as needed from the standard formats. Moreover, *WelQrate* dataset collection offers different split schemes, ensuring that it provides a comprehensive and standardized foundation for model evaluation and benchmarking. More discussion is found in Sec. 4.

### 3.1 Related Bioassays Identification and Data Retrieval

We elect to follow [13, 9] for bioassay identification, which have the following characteristics.

- **Data Relevance**: Selected targets are of therapeutic importance. Retrieved bioassays are relevant to the therapeutic target.
- **Data Quality & Reliance**: The experiment details described on PubChem are manually inspected by domain experts to ensure there are validation screens and established protocols and controls.
- **Data Consistency**: Selected bioassays are of the same unit of measurements (e.g. IC50) from the same experimental organization for a certain therapeutic target.

We then use the PubChem programmatic service[1] to retrieve all bioassays using their PubChem BioAssays Identifier (AID). The queried AID returns data containing PubChem Compound Identifiers (CIDs) for the small molecule compounds tested. Although PubChem claims that "For each BioAssay record, bioactivity data together with chemical structures (in isomeric SMILES format) are available for download"[2], some bioassays include non-isomeric SMILES (e.g., CID 124293). Therefore we use the PubChem Identifier Exchange Service [2] to retrieve isomeric SMILES from CID We also retrieve InChI with CID using the same method.

---

[1]https://pubchem.ncbi.nlm.nih.gov/docs/bioassays. Accessed in May 2024
[2]https://pubchem.ncbi.nlm.nih.gov/docs/identifier-exchange-service. Accessed May 2024

Tab. 1: Statistics of our 9 datasets in *WelQrate* dataset collection, which has coverage of various important drug targets, challenging but realistic low active percentages.

| Target Class | BioAssay ID (AID) | Target | Compound Type | Number of Compounds | Number of Actives | Percent Active | Unique BM Scaffolds |
|---|---|---|---|---|---|---|---|
| G Protein-Coupled Receptor (GPCR) | 435008* | Orexin 1 Receptor | Antagonist | 307,660 | 176 | 0.057% | 86,108 |
| | 1798 | M1 Muscarinic Receptor | Allosteric Agonist | 60,706 | 164 | 0.270% | 30,079 |
| | 435034 | M1 Muscarinic Receptor | Allosteric Antagonist | 60,359 | 78 | 0.129% | 39,909 |
| Ion Channel | 1843 | Potassium Ion Channel Kir2.1 | Inhibitor | 288,277 | 155 | 0.054% | 82,140C |
| | 2258 | KCNQ2 Potassium Channel | Potentiator | 289,068 | 247 | 0.085% | 82,247 |
| | 463087 | Cav3 T-type Calcium Channel | Inhibitor | 95,650 | 652 | 0.682% | 40,066 |
| Transporter | 488997* | Choline Transporter | Inhibitor | 288,564 | 236 | 0.082% | 82,343 |
| Kinase | 2689* | Serine Threonine Kinase 33 | Inhibitor | 304,475 | 120 | 0.039% | 85,314 |
| Enzyme | 485290 | Tyrosyl-DNA Phosphodiesterase | Inhibitor | 281,146 | 586 | 0.208% | 80,984 |

\* Indicates additional experimental measurements are available for those datasets. See Sec. 4.1 for details.

## 3.2 Data Processing

The retrieved compound data then undergoes the following processing steps.

- **Duplicate Removal**. Although the PubChem CID is theoretically a unique identifier for each compound, we found instances where the same compound had different CIDs (e.g., CID 130564 and CID 5311083). Therefore, we triple-check for duplicates using CID, isomeric SMILES, and InChI, respectively.

- **Hierarchical Curation**. In a typical HTS campaign, there are many bioassays, which fall into three categories. The primary screen is the initial screen for compound activity against a certain target, reducing the available compound library to a smaller set for further validation of activity. The activity threshold is typically set loosely to reduce the number of false negatives, resulting in a high false positive rate at this stage [9]. A confirmatory screen is a follow-up assay that validates the putative actives identified in the primary screen. A counter screen is set up to exclude compounds that show activity for an unwanted target, as a potential drug compound should have specificity for its intended target to reduce the chance of off-target toxicity.

  In our curation process, each bioassay undergoes manual inspection of the PubChem description to determine the relationships between assays and their experimental details. This manual inspection ensures that the bioassays are accurately linked in a step-by-step manner, forming a hierarchical structure of curation. Hierarchical curation involves organizing bioassays in levels, starting from primary screens, followed by confirmatory screens, and finally counter screens. This method reduces the false positive rate in the datasets by systematically validating and filtering the compounds through multiple layers of screening. A simple hierarchical curation example is shown in Fig. 2. All curation hierarchies can be found in the supplement Sec. 1.2.

- **Parser Filter.** We pass the molecules through RDKit [14] and inspect for any parsing errors.

- **Inorganic Substance Filter.** PubChem may contain inorganic substances, often present as counter ions resulting from the synthesis process rather than being part of the active component.

- **Handling of Mixtures**. The data from PubChem may include mixtures of multiple substances. We adapt the rules in [15] for handling mixtures: If the mixture is a duplicate of the same molecule, only one will be kept. If the molecular weight difference in the mixture is less than or equal to five, the mixture is discarded since it is hard to decide which molecules to keep. The remaining mixtures go through an inorganic filter and a druglikeness filter (Lipinski's Rule of Five is used as the druglikeness filter [16]) to retain only organic, druglike molecules.

- **Neutralization**. Data from PubChem may contain charged molecules, so we neutralize them using neutralize-by-atom algorithm [17].

- **Aromatization**. The original data contains the kekulized form of molecules (i.e., alternating double and single bonds in the aromatic system). Technically, the bonds should have equal properties; therefore, we convert the kekulized bonds into aromatic bonds.

- **PAINS Filter**. Pan-Assay Interference Compounds (PAINS) [18] tend to react non-specifically with numerous biological targets rather than specifically affecting one desired target and therefore need to be filtered out. There are three layers of promiscuity, optical interference, and other interference patterns. Details can be found in supplement Sec. 1.3.
- **Druglikeness Filter**. Druglikeness is assessed based on Lipinski's Rule of Five [16].
- **Expert Verification**. Although our pipeline is automated, flagged molecules with errors are still inspected to ensure the quality of our dataset. For instance, CID 409301 was excluded only after confirming the inconsistency between its InChI (indicating no covalent bonds) and SMILES (showing the presence of a covalent bond) representations.

## 3.3 Additional Data Format Generation

Additional data formats are provided for fair benchmarking. However, researchers are encouraged to generate their own SDF, 2D, and 3D Graphs, or any file formats they need from the standard formats. More discussion can be found in Sec. 4.1 and details are in supplement Sec. 1.4.

Corina [19, 20, 21] v5.0 is used to generate the SDF with a low energy 3D conformation. We note that a few molecules (123.4 on average per dataset) are filtered out by Corina during the generation of the 3D conformation, but none of these are active molecules, ensuring that datasets retain enough active signals. Graphs are designed using PyTorch Geometric [22], with atoms defined as nodes and 28-dimensional pre-defined node features (see supplement Sec. 1.4). The 2D Graph uses bond connectivity as edges and includes pre-defined edge attributes (see supplement Sec. 1.4). The 3D Graph defines edge existence if two nodes are within a certain 3D Euclidean distance. Following prior work [23], we use 6 angstroms as the distance cutoff to minimize the impact of molecular flexibility. Additionally, the 3D Graph contains *pos* as a node attribute to encode 3D coordinates.

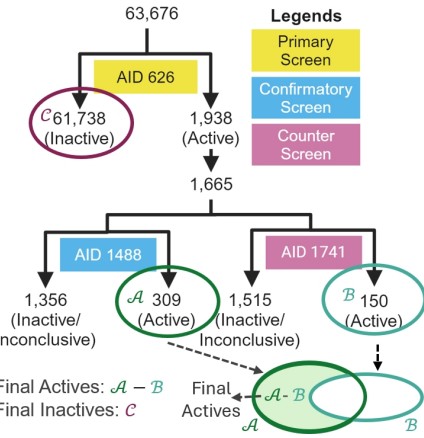

Fig. 2: An example of the hierarchical curation with AID 1798. Initially 63,676 compounds go through a primary screen (AID 626). The found 1,665 actives further go through a confirmatory screen (AID 1488) to verify their activities, and those showing activity in a counter screen (AID 1741) are excluded from the final active set.

# 4 *WelQrate*-Evaluation Framework

Currently, researchers in the field use varying methods for featurization, 3D conformation generation, and train/validation/test splitting. Moreover, existing datasets may not accurately reflect real-world drug discovery scenarios and often contain experimental artifacts, leading to noisy data. To ensure fair model comparison, we propose a standardized benchmarking protocol.

## 4.1 High-Quality Datasets for Fair Benchmarking

As detailed above, *WelQrate* dataset collection offers not only high-quality data from rigorous curation, but also additional data formats to facilitate standardized benchmarking. These additional formats include pre-defined atom and bond feature sets and pre-generated 3D conformations, establishing a common ground for fair model comparison. For those focused on model design, the standardized featurization we provide supports a consistent basis for evaluation. Nevertheless, we emphasize the importance of featurization for advancing the field and we strongly encourage researchers to innovate and benchmark novel features generated from the standard data formats. Similarly, while 3D conformations are provided, researchers are welcome to generate their own if relevant to their work.

The *WelQrate* dataset collection contains compounds labeled as active or inactive; however, some datasets also include additional experimental measurements that quantify activity values, available only for active molecules due to bioassay cost constraints. Researchers with drug discovery expertise could design a regression task to leverage this extra information. Further details on these datasets and a discussion are provided in the supplement Sec. 1.5.

Future updates to the *WelQrate* dataset collection, such as new targets, will be versioned and maintained at WelQrate.org, and we request that researchers report the version number used to ensure reproducibility.

## 4.2 Evaluation Metrics with Realistic Consideration

In the real-world drug discovery campaign, only the top-predicted molecules will be purchased or synthesized and those predicted to be inactive are of less concern [9]. A traditional evaluation metric for classification such as Receiver-Operating-Characteristic Area Under the Curve (AUC) is not ideal in this case, as it evaluates the model's overall performance for both actives and inactives Instead, we use four metrics that specifically focus on gauging the model's ability to correctly rank the active molecules in a high position in the list. A brief introduction of each metrics is shown below. More details of metrics can be found in the supplement Sec. 1.6.

- **logAUC**$_{[0.001,0.1]}$ measures logarithmic area under the receiver-operating-characteristic curve at false positive rates between [0.001, 0.1] [24, 25]. A perfect classifier gets a logAUC$_{[0.001,0.1]}$ of 1, and a random classifier gets a value of around 0.0215.

- **BEDROC** ranges from 0 to 1, where a score closer to 1 indicates better performance in recognizing active compounds early in the list [26].

- **EF**$_{100}$ measures how well a screening method can increase the proportion of active compounds in a selection set, compared to a random selection set [27]. Here we select the top 100 compounds as the selection set.

- **DCG**$_{100}$ aims to penalize a true active molecule appearing lower in the selection set by logarithmically reducing the relevance value proportional to the predicted rank of the compound within the top 100 predictions [28].

Researchers are also encouraged to incorporate additional metrics with realistic drug discovery considerations to strengthen model evaluation further.

## 4.3 Data Splitting for Robust Evaluations

We propose two standard dataset split methods for benchmarking: random and scaffold. Given that dataset splits significantly impact model performance, we recommend nested cross-validation as the ideal standard for random splits if resources allow, as it ensures robust evaluation. However, recognizing computational constraints, we also suggest an alternative approach that balances the rigor of nested cross-validation with the efficiency of a single test set, allowing all data points to serve as test sets at different stages to maximize robustness and minimize bias.

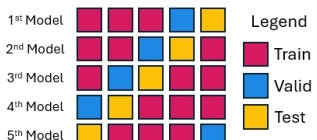

Fig. 3: Illustration of the adapted cross-valiation.

Traditional cross-validation often tests only a subset of data while tuning hyperparameters on the rest, leading to partial test coverage. Although nested cross-validation provides thorough testing, it requires substantial resources due to repeated model training across data splits. To set a practical standard for large-scale benchmarks, we propose a modified approach where all data is eventually tested, but we simplify the process by selecting only a single split at the inner level (the fold with the validation set immediately precedes the test set, Fig. 3). This strategy maintains a comprehensive test set while limiting the model count to five, enhancing computational efficiency without compromising robustness. See supplement Sec.2.2 for detailed discussion.

For scaffold splits, we propose a standard that supports scaffold hopping, a core task in drug discovery to create structurally novel compounds by modifying core structures, enhancing patentability, synthesis routes, and compound properties [29]. We recommend using Bemis-Murcko (BM) scaffolds [30] and a 3:1:1 training:validation ratio as a standardized benchmark, assigning any scaffold bin with more than 10% of the total molecules to the training set to ensure scaffold diversity across splits.

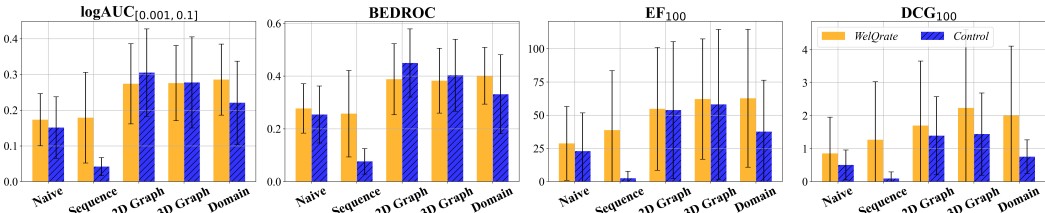

Fig. 4: Categorical performance comparison among different models (RQ1) trained respectively with WelQrate and Control Dataset (RQ2) (Note that individual model performances are shown in Fig. 6). Values are averages over performance across different datasets. Error bars denote standard error across multiple experimental runs and AIDs. For simplicity, *WelQrate* refers to *WelQrate* dataset collection in the legend.

## 5    Benchmarking

In this section, random split introduced in Sec. 4.3 is used for all experiments except for RQ4, in which scaffold split is used. All hyperparameters and training details are in the supplement Sec. 3.2 and Sec. 3.3.

### 5.1    RQ1: How Do Different Models And Data Representation Affect Performance?

We evaluated the performance of different molecular representation learning models on the *WelQrate* dataset collection, encompassing three primary categories:

- **Sequence-Based:** SMILES2Vec [31], TextCNN [32].
- **2D Graph-Based:** Graph Convolutional Neural Network (GCN) [33], Graph Isomorphism Network (GIN) [34], Graph Attention Network (GAT) [35].
- **3D Graph-Based:** SchNet [36], DimeNet++ [37], SphereNet [38].

Additionally, we included two baseline models to contrast traditional descriptor-based approaches with modern deep learning techniques.

- **Naive Baseline:** Atomic-level Pooling averages the atomic features as molecular representations.
- **Domain Baseline:** Molecular-level Descriptor, specifically, the BioChemical Library (BCL) [39] is utilized to extract a domain-driven descriptor set, such as signed 2D and 3D autocorrelations [23] (with details in the supplement Sec. 3.1).

The orange bars in Fig. 4 illustrate the performance of these models across four realistic metrics. Our first observation is a trend of increasing model performance with greater model complexity (i.e., from left to right) across all metrics for the three primary categories and the Naive Baseline. However, an exception is that the Domain Baseline performs the best except for $DCG_{100}$ while only utilizing a basic Multi-Layer Perceptron (MLP) due to the high-quality molecular-descriptors from BCL. The outperformance of the Domain Baseline model under this benchmarking signifies the further potential of fostering collaboration between the machine learning community and domain experts.

### 5.2    RQ2: How Does Dataset Quality Impact Model Evaluation?

To examine the impact of dataset quality on model evaluation, we created a control dataset that includes only data directly from primary screens, bypassing the rigorous processing steps described in Section 3.2. This control dataset allows us to assess the significance of our curation pipeline. To ensure a fair comparison, we maintained identical test sets between the control and *WelQrate* dataset collection and only vary the training and validation sets used to train the models and tune the hyperparameters.

In Fig. 4 we present the performance of models trained on the control (blue) and WelQrate (orange). The first observation is that for the Naive Baseline and Sequence-Based methods there is a significant decrease in performance across all four metrics when trained on the control dataset. However, the 2D and 3D Graph-Based models trained on the control dataset actually outperform in $logAUC_{[0.001,0.1]}$

and BEDROC, but perform worse for $EF_{100}$ and $DCG_{100}$. We hypothesize this is due to the range of top selected candidates these metrics use for evaluation and suggest future work to dive deeper to better understand these differences. As for the Domain Baseline, the models trained on *WelQrate* dataset collection better in terms of all metrics. Overall, these findings align with data-centric AI [40, 41], highlighting the importance of dataset quality in training the model, but also the need for comprehensive evaluation metrics, as we observe the trends can vary across metrics showcasing the limitations of only leveraging a single metric.

### 5.3 RQ3: How Significant Is Featurization in Model Evaluation?

To investigate the impact of featurization on model evaluation, we created a dataset with commonly used one-hot encoding for atom types and compared it with the pre-generated features in the *WelQrate* dataset collection. For simplicity, this experiment is carried out with a small dataset AID 1798. Specifically, in this experiment, we excluded the Domain Baseline as it inherently cannot be converted to using one-hot atomic encodings, since it uses molecular-level descriptors. Sequence-based models were also excluded because they do not utilize atomic features.

Fig. 5 illustrates the comparison of model performance using one-hot encoding and predefined features. Although we only show one metric for space considerations, other metrics exhibit the same trend (and in the supplementary materials). The results indicate that models using one-hot encoding generally underperform compared to those utilizing predefined features. One strong exception to this is the Naive Baseline model where is remains quite stable. Overall, these results along with the strong performance of Domain Baseline (in RQ1) underscore the importance of advanced featurization techniques (i.e., data-centric AI [40, 42]) to enhance overall model performance.

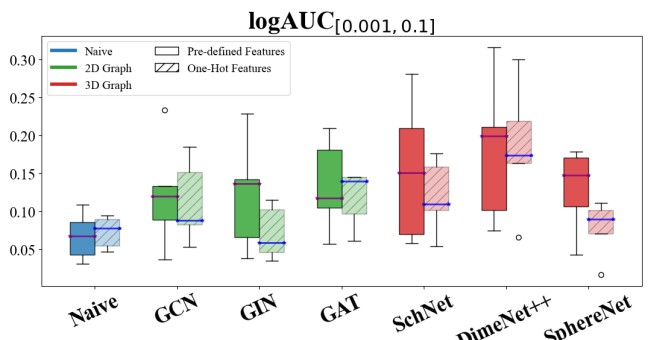

Fig. 5: Comparison of model performance using one-hot encoding and pre-defined features in *WelQrate* dataset collection (RQ3). Error bars denote standard error across multiple experimental runs.

We advocate for the research community to develop better featurization methods. However, it is imperative to recognize that innovative model architectures should be benchmarked using the same featurization (when possible) to ensure fair and accurate comparisons.

### 5.4 RQ4: How Do Different Models Perform Under Scaffold Splitting?

Fig. 6 shows that all models exhibit decreased performance under the scaffold split scenario, which aligns with our expectations. Additionally, another interesting finding is that while we observe a positive correlation between model complexity and performance earlier (i.e., Fig. 4), here the benefits of more advanced 2D and 3D Graph-Based deep learning models tend to disappear in the scaffold split scenario compared to random split; we hypothesize this could be related to an overfitting issue in these complex models. In fact, predicting the activity of molecules with unseen scaffolds represents a distribution-shift problem and is inherently more challenging for all models. Notably, the Domain Baseline model, without sophisticated architecture but with domain expert crafted descriptors, remains robust across all metrics, underscoring the critical importance of domain knowledge and featurization for scaffold hopping.

These observations highlight the necessity of developing models that can effectively handle the distribution shift associated with scaffold hopping. Future research should focus on enhancing the robustness of models to scaffold diversity and improving featurization techniques to better capture the underlying chemical properties crucial for accurate activity prediction.

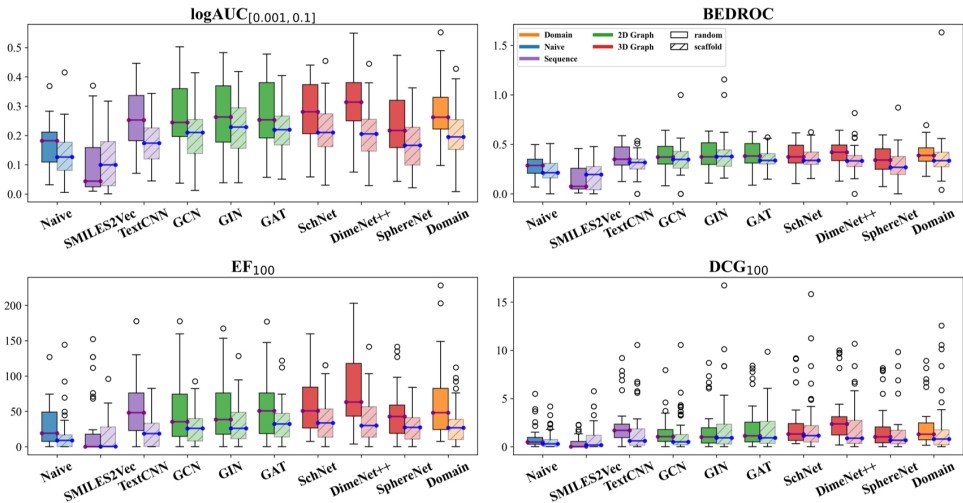

Fig. 6: Comparison of model performance under random and scaffold split (RQ4). Error bars denote standard error across multiple experimental runs and AIDs.

## 6 Limitations and Future Directions

Despite the advancements presented in this study, several limitations and future directions remain. First, given that the *WelQrate* dataset collection reflects real-world drug discovery scenarios having a low percentage of active compounds, this yields highly imbalanced datasets that pose challenges for off-the-shelf deep learning approaches. Thus, we encourage future research dedicated to class-imbalanced learning on graphs [43]. Additionally, the poor performance of models under scaffold splitting underscores the need for robust models capable of handling distribution shifts. Expanding the evaluation framework to include metrics such as ADMET (Absorption, Distribution, Metabolism, and Excretion) properties [44] could provide a more comprehensive assessment of model efficacy. Moreover, The current *WelQrate* version recommends using a standard 3D conformation, assuming the molecule is at its lowest energy state, generated by Corina. However, many methods, such as ETKDG used in RDKit [14], can be used to generate conformations. Future versions of WelQrate could expand beyond standard conformation to likely binding conformations. Future work could also explore using the datasets for generative tasks and incorporating domain knowledge to design better models and featurization techniques. Addressing these limitations will further advance AI-driven drug discovery, leading to more reliable and effective therapeutic solutions.

## 7 Conclusion

In this study, we introduced *WelQrate*, a new gold standard for benchmarking in small molecule drug discovery. Our contributions include rigorous data curation, a standardized evaluation framework, and extensive benchmarking of existing deep learning architectures. Through expert-designed curation pipelines, *WelQrate* dataset collection addresses prevalent issues, such as inconsistent chemical representations and noisy experimental data, ensuring high-quality labeling of active molecules crucial for reliable model training and evaluation. Our proposed evaluation framework encompasses critical aspects such as featurization, 3D structure generation, relevant evaluation metrics, etc., providing a reliable basis for model comparison and facilitating accurate and realistic evaluations in virtual screening tasks. Benchmarking experiments with *WelQrate* demonstrates how model performance is influenced by key factors such as model type, dataset quality, featurization, and data split schemes. By examining these aspects, we highlight the importance of each in achieving robust and reliable model evaluation, offering insights that can guide future developments in model selection and benchmarking standards in drug discovery.

The *WelQrate* dataset collection, along with detailed curation procedures and experiment scripts, is publicly available and maintained at WelQrate.org. We recommend broader adoption of these practices to set a new benchmark standard, ensuring more consistent and meaningful advancements in the field of drug discovery.

## Acknowledgments and Disclosure of Funding

The authors thank the members of Therapeutics Data Commons (TDC) for the discussion. The authors thank Joshua Bauer at the Institute of Chemical Biology, VU, and Alex Waterson at the Pharmacology Dept, VU, for their experience and guidance on the curation procedure. The authors thank Fabian Liessmann and Claiborne (Clay) Tydings for their insights and suggestions on improving the datasets. The authors thank Xiaohan Kuang for the help with hardware support.

Y.L acknowledges the Nvidia Hardware Grant for providing the A6000 for accelerating the benchmarking.

Work in the Meiler laboratory is supported through NIH (R01 GM080403, R01 HL122010, R01 DA046138). J.M. is supported by a Humboldt Professorship of the Alexander von Humboldt Foundation. J.M. acknowledges funding by the Deutsche Forschungsgemeinschaft (DFG, German Research Foundation) through SFB1423, project number 421152132 and through SPP 2363 for financial support. J.M. is further supported by Federal Ministry of Education and Research (BMBF) through the Center for Scalable Data Analytics and Artificial Intelligence (ScaDS.AI). This work is partly supported by BMBF (Federal Ministry of Education and Research) through DAAD project 57616814 (SECAI, School of Embedded Composite AI).

B.B. was supported by the National Science Foundation under grant 1763966.

This work in the Network and Data Science laboratory is partly supported by the National Science Foundation under grant IIS2239881.

Certain figures in the paper are created with BioRender.com.

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
