# Supplement
# WelQrate: Defining the Gold Standard in Small Molecule Drug Discovery Benchmarking

## Table of Contents

38th Conference on Neural Information Processing Systems (NeurIPS 2024) Track on Datasets and Benchmarks.

# 1 Datasets and Data Curation Pipeline

## 1.1 Limitations of Existing Datasets

One example of problem with MoleculeNet [57] is the SIDER dataset [29]. SIDER is a dataset for predicting side effect from the small molecule structure. It contains 27 classification tasks, corresponding to the 27 system organ classes following MedDRA classifications [1]. If taking a closer look at the MedDRA classification on the system organ level on its website, we can find a claim of "System Organ Classes (SOCs) which are groupings by aetiology (e.g. Infections and infestations), manifestation site (e.g. Gastrointestinal disorders) or purpose (e.g. Surgical and medical procedures). In addition, there is a SOC to contain issues pertaining to products and one to contain social circumstances."[1]. This means, not all system organ classes are disease related, but could be related to product or social circumstances, which is irrelevant to small molecule structures. In fact, the two tasks among the 27 tasks are named "Social circumstances" and "Product issues", that corresponds to the claims above. Predicting such label from molecular structure alone is futile and therefore does not serve the purpose of a benchmarking dataset. The other problematic example in MoleculeNet is the PCBA dataset, originally used in [44]. However, as claimed in the original paper, "It should be noted that we did not perform any preprocessing of our datasets, such as removing potential experimental artifacts". And we have demonstrated the importance of removing the experimental artifacts in the data processing pipeline in the main text. There are more example issues with MoleculeNet that can be found in [52].

For Therapeutics Data Commons (TDC) [24], we used filters in our pipeline on small molecule-related tasks on and found issues with them. Tab. 1 lists the details.

Tab. 1: Examples of issues found with our filters in small molecule & drug discovery related datasets in TDC. The promiscuity filter is not applied due to the long running time. Note that some datasets display significant large ratio of actives not passing (ANP) the filters, compared to their total number of actives, such as HIV.

| Type | Dataset | # Actives | # Compounds | # Duplicates | # Inorganic Compounds | # Mixtures | #ANP Optical Interference Filter | #ANP Other Interference Patterns Filter |
|---|---|---|---|---|---|---|---|---|
| HTS | SARSCoV2_Vitro_Touret | 88 | 1484 | 4 | 0 | 586 | 3 | 4 |
| HTS | SARSCoV2_3CLPro_Diamond | 78 | 880 | 1 | 1 | 15 | 0 | 2 |
| HTS | HIV* | 1443 | 41127 | 0 | 55 | 3087 | 0 | 294 |
| Tox | hERG | 451 | 655 | 7 | 0 | 12 | 7 | 17 |
| Tox | hERG_Karim | 6718 | 13445 | 0 | 0 | 301 | 11 | 212 |
| ADME | Pgp_Broccatelli | 650 | 1218 | 6 | 0 | 0 | 8 | 63 |
| ADME | CYP2C19_Veith | 5819 | 12665 | 0 | 8 | 521 | 199 | 318 |
| ADME | CYP2D6_Veith | 2514 | 13130 | 0 | 7 | 549 | 134 | 126 |
| ADME | CYP3A4_Veith | 5110 | 12328 | 0 | 6 | 549 | 245 | 193 |
| ADME | CYP1A2_Veith | 5829 | 12579 | 0 | 6 | 542 | 498 | 333 |
| ADME | CYP2C9_Veith | 4045 | 12092 | 0 | 7 | 538 | 167 | 236 |
| Tox | Tox21* | 309 | 7265 | 0 | 113 | 233 | 5 | 18 |

*These datasets appear in MoleculeNet as well.

As mentioned in the introduction in the main paper, there are also issues with inconsistent representations and undefined stereochemistry. We list an example for each in Fig. 1 and Fig. 2.

---

[1]https://www.meddra.org/how-to-use/basics/hierarchy, Accessed Jun 2024

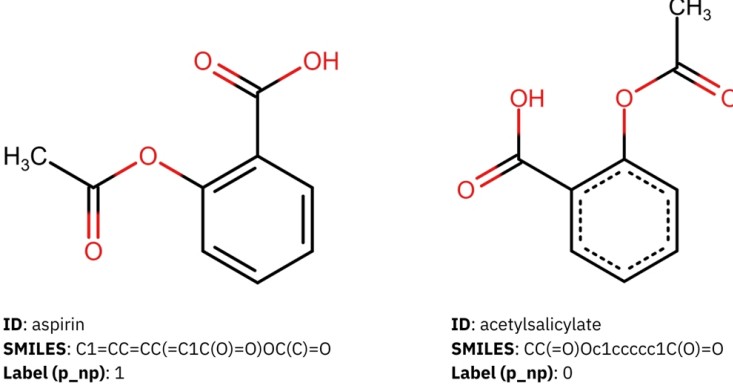

**ID**: aspirin
**SMILES**: C1=CC=CC(=C1C(O)=O)OC(C)=O
**Label (p_np)**: 1

**ID**: acetylsalicylate
**SMILES**: CC(=O)Oc1ccccc1C(O)=O
**Label (p_np)**: 0

Fig. 1: An example of inconsistent molecular representations for the same molecule found in MoleculeNet's BBBP dataset. These two should be the same molecule (Acetylsalicylate is known as Aspirin) but the left one is represented as kekulized form while the right one in aromatic form. Moreover, these two have different labels.

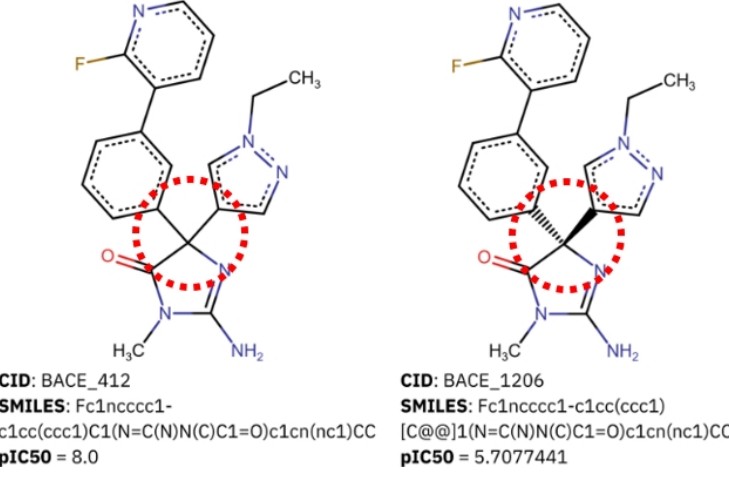

**CID**: BACE_412
**SMILES**: Fc1ncccc1-c1cc(ccc1)C1(N=C(N)N(C)C1=O)c1cn(nc1)CC
**pIC50** = 8.0

**CID**: BACE_1206
**SMILES**: Fc1ncccc1-c1cc(ccc1)[C@@]1(N=C(N)N(C)C1=O)c1cn(nc1)CC
**pIC50** = 5.7077441

Fig. 2: An example of undefined stereochemistry MoleculeNet's BACE dataset. The highlighted circle marks a chiral center, which have several possible 3D arrangements. On the left, it is not specified how it is arranged (undefined stereochemistry). On the right, it is specified ( the wedge denotes the bond is pointing outward, while the dash denotes the bond pointing inwards). This lack of definition can lead to ambiguity, as different stereoisomers of the same molecule can have vastly different biological activities. If stereochemistry is not explicitly defined, the model may incorrectly assume that different stereoisomers are the same compound, potentially leading to erroneous predictions.

## 1.2 Dataset Description & Hierarchical Curation Details

In a typical HTS campaign, there are many bioassays, which fall into three categories. The primary screen is the initial screen for compound activity against a certain target, reducing the available compound library to a smaller set for further validation of activity. The activity threshold is typically set loosely to reduce the number of false negatives, resulting in a high false positive rate at this stage [11]. A confirmatory screen is a follow-up assay that validates the putative actives identified in the primary screen. A counter screen is set up to exclude compounds that show activity for an unwanted target, as a potential drug compound should have specificity for its intended target to reduce the chance of off-target toxicity.

In our curation process, each bioassay undergoes manual inspection of the PubChem description to determine the relationships between assays and their experimental details. This manual inspection ensures that the bioassays are accurately linked in a step-by-step manner, forming a hierarchical structure of curation. Hierarchical curation involves organizing bioassays in levels, starting from primary screens, followed by confirmatory screens, and finally counter screens. This method reduces the false positive rate in the datasets by systematically validating and filtering the compounds through multiple layers of screening.

In the following diagrams, the AIDs can be used to look up the original bioassays on PubChem. Please note that the number of compounds shown in the diagrams are before other filters (e.g., PAINS filter) are applied.

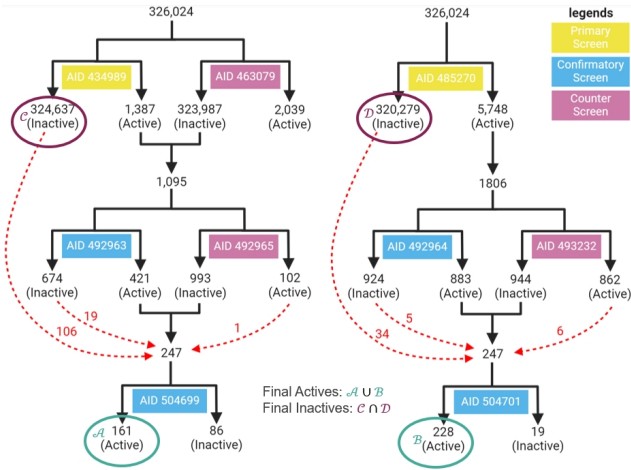

Fig. 3: Hierarchical curation pipeline for AID435008

## AID435008

This dataset focused on identifying Orexin receptor type 1 (OX1R) antagonists. Through its selective interaction with neuropeptide orexin A, OX1R demonstrates critical functions in feeding [21], sleep [16, 21], mood [2], and addiction [50]. Two primary screens, AID434989 and AID485270, identified antagonists of OX1R through fluorescence-based cell-based assay and FRET-based cell-based assay technologies. AID463079 countered AID434989 through a similar fluorescence-based assay that evaluated non-selectively inhibition of Gq signaling in the parental CHO cell line (without GPCR transfection). Compounds that were active in the primary screen AID434989 and inactive in the counter screen AID463079 were subjected to a second layer of confirmatory screen AID492963 and counter screen AID492965, both using fluorescence-based cell-based assays with conditions similar to AID434989. Compounds that were active in the primary screen AID485270 were confirmed by the confirmatory screen AID492964 and tested for selectivity by the counter screen AID493232, both being Homogeneous Time Resolved Fluorescence (HTRF)-based cell-based assays. Actives from the confirmatory screen AID492963 that were inactive in AID492965 were confirmed by the last layer of confirmatory screens AID504699, while actives from AID492964 that were inactive in AID493232 were confirmed by AID504701. The two confirmatory screens in this last layer were both dose-response assays.

Since the two experimental pipelines used two separate technologies in identifying OX1R antagonists, the final actives were taken from the union of the active subsets from the last two confirmatory screens, AID504699 ($\mathcal{A}$ in Fig. 3), and AID504701 ($\mathcal{B}$ in Fig. 3). The final inactive compounds were taken from the intersection of the two inactive subsets of the two primary screens, AID434989 ($\mathcal{C}$ in Fig. 3), and AID485270, ($\mathcal{D}$ in Fig. 3) to reduce false-negative rate, since we found some inactive compounds in one screen that were active in the other. Note that we also found some compounds that were not strictly aligned with the assay descriptions (*red, dashed arrows*), i.e., being inactive in previous primary and confirmatory screens or active in the counter screens but still included in the follow-up confirmatory screens. Such occurrences are often due to the fact that experimentalists might decide to move on with compounds whose activities were found to be close to the threshold defined for active compounds or sharing common scaffolds with confirmed active compounds. Therefore, active readouts from such compounds were kept in the final active set, and inactive ones were included in the final inactives. This practice reduced both false-positive and false-negative rates and enabled more rigorous training tasks by ensuring that similar scaffolds were present in both classification labels.

Dose-response data as the additional experimental measures (IC50) were averaged from the readouts of AID504699 and AID504701 for active compounds. For inactive compounds whose IC50 values were not available, an arbitrarily high IC50 of 1000 $\mu$M was chosen. See Sec. 1.5 for details.

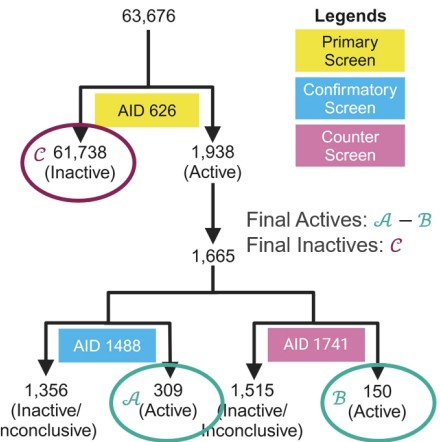

Fig. 4: Hierarchical curation pipeline for AID1798

**AID1798**

The $G_q$-coupled GPCR M1 Muscarinic Receptor is a seven-transmembrane domain receptor that plays a significant role in cognitive function and has been targeted for therapeutic purposes, particularly in treating Alzheimer's Disease and schizophrenia [28, 35, 10, 6]. This dataset focuses on identifying small agonists for M1 receptors (Fig. 4), which potentially improve cognitive performance and reduce psychotic symptoms. The primary screen AID626 used a cell-based fluorometric calcium assay to discover positive allosteric modulators of the M1 muscarinic receptor. Actives from this assay were further confirmed by the confirmatory screen AID1488 and tested for selectivity by the counter screen AID1741, which evaluated cross-activity with the closely-related M4 muscarinic receptor.

The final active set included compounds that were active in AID1488 ($\mathcal{A}$ in Fig. 4) without demonstrated activity in AID1741 ($\mathcal{B}$ in Fig. 4). The final inactive set was retrieved from inactive readouts in the primary screen AID626 ($\mathcal{C}$ in Fig. 4).

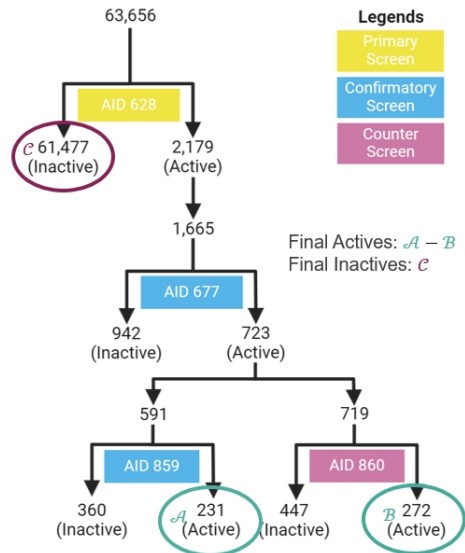

Fig. 5: Hierarchical curation pipeline for AID435034

**AID435034**

This dataset presents the HTS result from a collection of 4 HTS BioAssays aiming at identifying antagonists of the M1 Muscarinic Receptor, relevant to treatments of form-deprivation myopia [4] and impaired visual recognition [56]. The primary screen AID628 shared the same tested compound set with the assay AID626 (refer to the AID1798 dataset description above). Potential antagonists identified by AID628 were tested by the confirmatory screen AID677 through a cell-based fluorometric calcium assay. Actives from AID677 were subjected to another layer of a confirmatory screen (AID859, with the same condition with AID677) and a counter screen (AID860, with similar assay condition but for non-specific activity against the M4 muscarinic receptor).

Final active set included active compounds from AID859 ($\mathcal{A}$ in Fig. 5), subtracted by non-specific binders revealed in AID860 ($\mathcal{B}$ in Fig. 5). Final inactive set was taken from the inactive subset of the primary screen AID628 ($\mathcal{C}$ in Fig. 5).

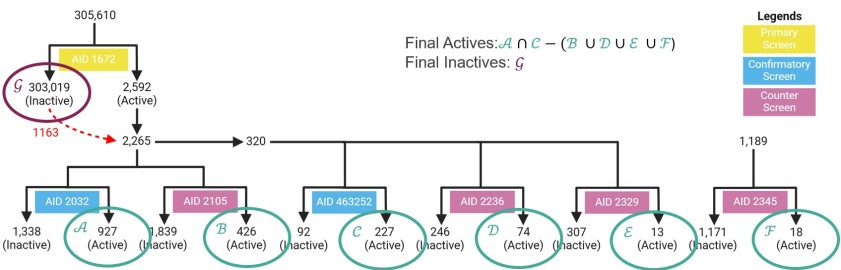

Fig. 6: Hierarchical curation pipeline for AID1843

## AID1843

This dataset focused on developing small molecular inhibitors to the Inward-rectifying Potassium Ion Channel Kir2.1 (IRK-1) (Fig. 6). Modulators of the Kir2.1 channel serve as leads for developing anti-arrhythmic drugs, including treatments of short- and long-QT syndromes, Andersen syndrome, and a range of other cardiovascular, neurological, renal and metabolic disorders [15, 51, 59]. The primary screen AID1672 identified compounds that inhibit/block IRK-1 using a thallium assay on the IRK-1-expressed HEK293 cell line. Actives from this assay were confirmed by the confirmatory screen AID2032 under the same assay condition as AID1672. A subset of 320 active compounds from AID1672 were confirmed by the confirmatory screen AID463252 with high precision with an automated population patch-clamp electrophysiology assay. A series of counter screens, AID 2105 (parental HEK293 cells), AID 2236 (hERG CHO cells), AID 2329 (KCNK9-expressed HEK293 cells), and AID2345 were performed to identify active compounds exhibiting non-specific binding effects against IRK-1. The last counter screen AID2345 was retrieved from a separate project, in which active compounds against KCNQ2 channels from AID2239 (see the section on AID2258, dataset for KCNQ2 Potassium Channels) were tested for activity against IRK-1. Compounds returning active readouts in AID2345, therefore, should have non-specific activity against IRK-1 and need to be discarded from the final actives. Among tested compounds in AID2032, we found 1163 compounds that were inactive in the primary screen AID1672 (*red, dashed arrow*). However, all of them returned inactive readouts and should be included in the final inactive set. Similarly, 33 inactives from AID1672 were tested in AID463252 but did not demonstrate activity against IRK-1.

Despite its significantly smaller size, the confirmatory screen AID463252 has a much more stringent assay condition than AID2032, given its patch-clamp method. Therefore, final actives were taken from the intersection of the active subset from this dataset ($\mathcal{C}$ in Fig. 6) and active compounds in AID2032 ($\mathcal{A}$ in Fig. 6), removing any compounds that demonstrated non-specific activity against IRK-1 from the counter screens (union of $\mathcal{B}, \mathcal{D}, \mathcal{E}$ and $\mathcal{F}$ in Fig. 6). The final inactive set was taken from the inactive subset of the primary screen ($\mathcal{G}$ in Fig. 6).

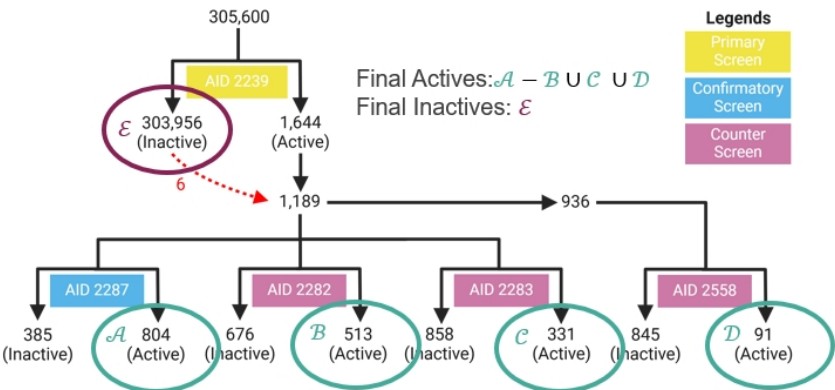

Fig. 7: Hierarchical curation pipeline for AID2258

## AID2258

This dataset presents the HTS result from a collection of 5 HTS BioAssays aiming at identifying potentiators of KCNQ2 Potassium Channels (Fig. 5). This channel belongs to the Kv7 voltage-gated potassium channel family, important for the regulation of neural excitability and resting state of cells [7, 19]. Dysregulation of KCNQ2 was shown to be involved in severe neonatal-onset developmental and epileptic encephalopathy [36, 25]. The primary screen AID2239 identified potentiators of the KCNQ2 potassium channel that caused an increase in fluorescent signal intensity measured by a thallium assay. The confirmatory screen AID2287 tested actives from the primary screen under similar conditions in duplicate. Three counter screens were included in our dataset, identifying non-selective compounds showing response for CHO-K1 cell activity (AID2282), non-specific effects on KCNQ1 (AID2283), and response in KCNQ2-W236L-CHO cells (AID2558).

The final active set included active compounds in AID2287 ($\mathcal{A}$ in Fig. 7), subtracted by compounds that demonstrated activity in any of the three counter screens (union of $\mathcal{B}, \mathcal{C}$ and $\mathcal{D}$ in Fig. 7). The final inactive set was taken from the inactive subset of the primary screen ($\mathcal{E}$ in Fig. 7). Six inactive compounds in the primary screen AID2239 were found to be re-tested in the follow-up screens (*red, dashed arrow*). Final active readouts among these six compounds caused their exclusion from the final inactive set.

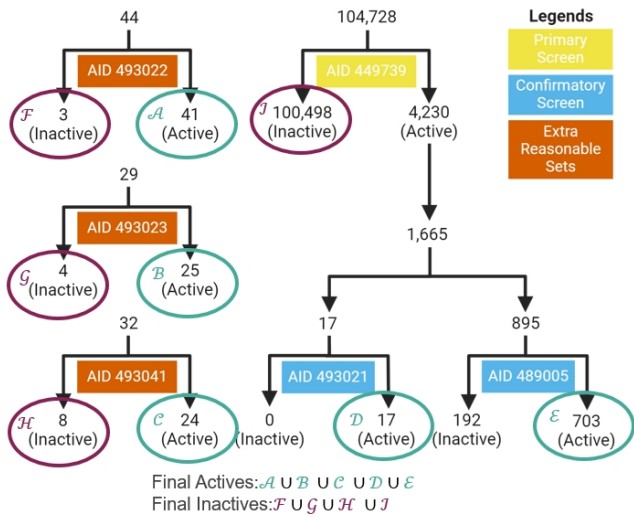

Fig. 8: Hierarchical curation pipeline for AID463087

**AID463087**

This dataset was compiled from 6 BioAssays aiming at finding inhibitors of Cav3 T-type Calcium Channels. The Cav3 Transient-type (T-type) calcium channel is a T-type low voltage-activated calcium channel expressed throughout the nervous system and is involved in various physiological functions, such as modulating neuronal firing patterns [23, 26]. It serves as an attractive target for treating chronic pain [12], epilepsy, and pulmonary hypertension [39, 42]. The primary screen AID449739 identified inhibitors of Cav3 T-type calcium channels by measuring calcium fluorescence modulation in a Cav3.2 expressing cell line. We selected five follow-up dose-response assays, AID489005, AID493021, AID493022, AID493023, and AID493041 to include in this dataset. While AID489005 employed the same assay as AID449739, the other four dose-response assays applied similar conditions to the primary Screen but were tested in 11-point 3-fold CRC experiments in triplicate. Note that because AID493022, AID493023, and AID 493041 tested compounds synthesized at Vanderbilt, they do not share any compound with the primary screen (whose tested compounds came from the small molecule library provided by the Molecular Libraries Small Molecule Repository). Therefore, these assays were referred to as *Extra Reasonable Sets*, and only AID489005 and AID493021 were referred to as confirmatory screens.

As shown in Fig. 8, the final set of active compounds was acquired by taking the union of all confirmatory screens ($\mathcal{D}, \mathcal{E}$ in Fig. 8) and extra reasonable sets ($\mathcal{A}, \mathcal{B}$ and $\mathcal{C}$ in Fig. 8). The final inactive set included the inactive subset of the primary screens ($\mathcal{I}$ in Fig. 8) and all inactive compounds in the extra reasonable sets ($\mathcal{F}, \mathcal{G}$ and $\mathcal{H}$ in Fig. 8).

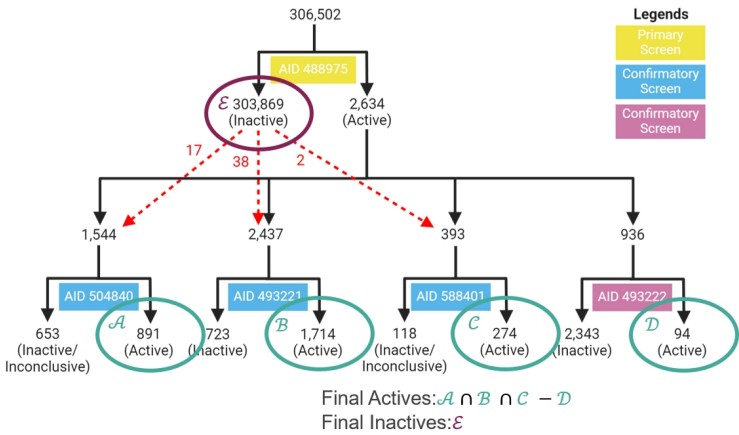

Fig. 9: Hierarchical curation pipeline for AID488997

### AID488997

This dataset presents the HTS result from a collection of 5 HTS BioAssays aiming at identifying inhibitors of the human high-affinity Choline Transporter (CHT). Acetylcholine (ACh) has vital modulatory functions over arousal, motor and cognition [43, 20], and its pathway was found to be vulnerable in certain neurological disorders such as Alzheimer's Disease (AD) [55] . Without the capability to synthesize choline *de novo*, cholinergic neurons rely on choline uptake for ACh synthesis through the high-affinity CHT [40]. CHT therefore is suggested as a promising target for cholinergic therapies in AD and other diseases whose pathology is regulated by cholinergic signaling. The Primary Screen AID488975 identified compounds that inhibit the choline-induced membrane depolarization by CHT in the CHT-expressing HEK293 cell line through a choline-induced membrane potential assay. Active compounds from AID488975 were confirmed by a series of confirmatory Screens, among which the three confirmatory Screens, AID504840, AID493221, and AID588401, were chosen for their significantly larger sizes than the other screens. These screens employed the same conditions as presented in the Primary Screen in duplicate (for AID493221) or similar but with 5-point concentration response curve (CRC) (for AID504840) or 10-point CRC (for AID588401). The counter screen AID493222 evaluated active compounds from AID488975 for non-specific activity in the parental HEK293 cell line.

As shown in Fig. 9, the final active set included the intersection of active subsets in AID504840 ($\mathcal{A}$ in Fig. 9), AID493221 ($\mathcal{B}$ in Fig. 9), and AID588401 ($\mathcal{C}$ in Fig. 9), removing any non-specific compound that was active in AID493222 ($\mathcal{D}$ in Fig. 9). Compounds found inactive in AID488975 ($\mathcal{E}$ in Fig. 9) were included in the final inactive set. In addition, we found 17, 38, and 2 inactive compounds from AID488975 (*red, dashed arrows*) that were tested in AID504840, AID493221, and AID588401, respectively. Among those, some were found to be active in these confirmatory screens. Since the conditions of confirmatory screens were more stringent than the primary screen, we decided not to remove these compounds from our dataset and include any inactive readout from these compounds in our final inactive set.

Dose-response as the additional experimental measures (IC50) was taken from the average IC50 values reported in the two assays AID504840 and AID588401. For inactive compounds, an arbitrarily high IC50 value of $1000\mu$M was chosen. See Sec. 1.5 for details.

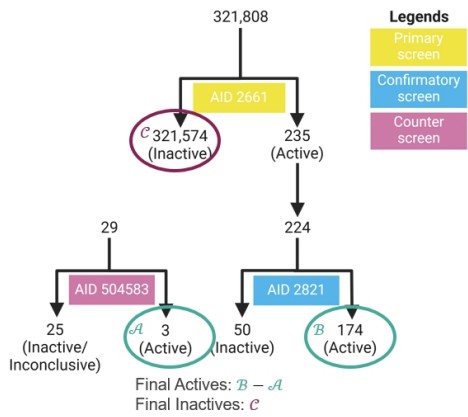

Fig. 10: Hierarchical curation pipeline for AID2689

**AID2689**

The goal of this project is to identify inhibitors for Serine/Threonine Kinase 33 (STK33), a pro-tumor protein involved in mitotic DNA damage checkpoint and protein autophosphorylation and shown to be important for survival and proliferation of mutant KRAS-dependent cancer cells [46, 8]. The primary screen AID2661 identified inhibitors of STK33 by measuring the change in luminescent signal, an indication of kinase activity of purified STK33 when preincubated with potential inhibitors (cell-free assays). The confirmatory screen AID2821 re-tested actives from AID2661 by a dose-response assay similar in experimental conditions. The counter screen AID504583 evaluated a subset of compounds for STK33 selectivity by measuring Protein Kinase A inhibition.

The final active set included compounds that were found active in AID2821 ($\mathcal{B}$ in Fig. 10), removing any active compounds in the counter screen ($\mathcal{A}$ in Fig. 10). The final inactives were taken from the inactive subset of the primary screen ($\mathcal{C}$ in Fig. 10).

The additional experimental measures data (EC50) was extracted from AID2821 for final active compounds. Since the EC50 ($\mu$M) values reported for actives ranged from 0.100 to 120.0, an arbitrarily high EC50 value of 1000 $\mu$M was chosen for inactive compounds. See Sec. 1.5 for details.

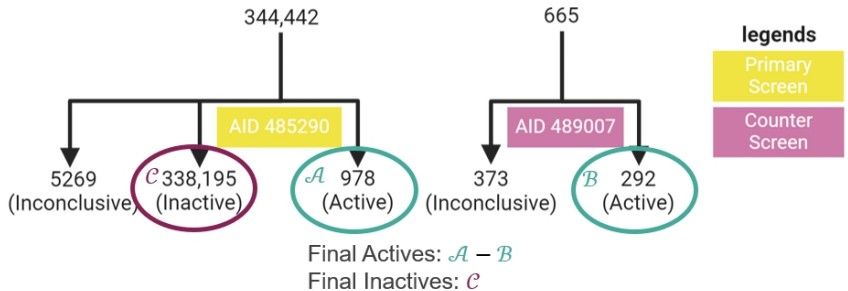

Fig. 11: Hierarchical curation pipeline for AID485290

**AID485290**

The Human tyrosyl-DNA phosphodiesterase 1 (TDP1) is involved in the repair of DNA lesions caused by topoisomerase 1 (Top1) inhibitors, which are used in chemotherapy. This dataset compiled HTS screenings focused on developing small molecular inhibitors for TDP1, which potentially enhance the cytotoxic effects of Top1 inhibitors, making cancer cells more susceptible to chemotherapy [32, 3]. The primary screen AID485290 identified inhibitors of TDP1. The follow-up counter screen AID489007 used the AlphaScreen (AS) detection method measuring the loss in AS signal due to the cleavage of the bond between TDP1's oligonucleotidic DNA substrate and fluorescein isothiocyanate.

The final active set contained any active compound in AID485290 ($\mathcal{A}$ in Fig. 11), excluded by actives in AID489007 ($\mathcal{B}$ in Fig. 11). The final inactive set included inactives in AID485290 ($\mathcal{C}$ in Fig. 11).

## 1.3 Data Processing - PAINS Filters

There are three layers of filters employed to detect PAINS: Promiscuity Filter, Optical Interference, and Other Interference Patterns Filter. Details of them can be seen below.

**Promiscuity Filter**

Promiscuous compounds refer to compounds that demonstrate activity against multiple, often unrelated, targets. These compounds hold less interest in drug discovery due to their potential to trigger unintended effects and adverse interactions [58]. [45] establishes Frequency of Hits (FoH), the ratio of the number of assays in which a compound is tested active and the number of assays in which it was tested, as a measure of promiscuity. A FoH larger than 0.26 is considered a potential nonspecific binder. FoHs for all compounds in our datasets were calculated through the following steps adapted from [45]:

- For each compound, the information on its tested assays was retrieved from PubChem's FTP record[2].
- For each of the assays tested, the sequence of the protein target was retrieved by the Biopython API [14] to the Entrez database of NCBI [54].
- Given all sequences of the proteins tested for each compound, a Multiple Sequence Alignment was performed to find the Percent Sequence Identity (%SI) between these proteins. The Clustal Omega [47] server provided by the EMBL-EBI Job Dispatcher framework [31] was used for this purpose.
- The general rule is that an assay whose target is highly similar to other targets should contribute less significantly to a compound's promiscuity. Therefore, the percentage identities were used to calculate the weight of each assay: w = 1 - %SI/100.
- FoH for each compound was calculated by the formula: FoH = wACC/nTAC, with wACC being the weighted total number of assays tested where the compound was identified as active. nTAC is the total number of assays tested, normalized by weights. Only assays with over 10,000 compounds tested were considered.
- Compounds with FoH over 0.26 were discarded.

**Optical Interference Filter** Recent HTS efforts have been done to identify frequent hitters that cause false-positive signals due to their chromo/fluorogenic properties [48]. In this benchmark, we filter potential fluorescent interferences by eliminating compounds that were experimentally confirmed by autofluorescent assays, including AID 587, AID 588, AID 590, AID 591, AID 592, AID 593, and AID 594.

**Other Interference Patterns Filter** PAINS compounds were identified in the rest of our data through a SMARTS substructural matching filter provided by RDkit [30]. This substructural filter includes a catalog of SMARTS versions of compounds previously identified as PAINS from a number of protein-protein interaction screens using the AlphaScreen technology [5]. Any compound that returns a True flag from this filter was discarded.

## 1.4 Additional Format Details

### 1.4.1 SDF Generation

The resulting .txt data files from the curation were converted into SDF files by Corina Classic [13] v5. 3D conformations of molecules were generated from input SMILES strings with missing hydrogens added (driver option -d wh) and aromatic bonds explicitly included as *4* in the output file (ouput option -o mdlbond4). PubChem CID for each compound was included in the title line, and the activity label for each compound (Active/Inactive) was included in the comment line of the SDF format. An example of converting an input.txt file into an output.sdf file is shown below:

```
corina5 -i t=smiles,sep=";",scn=2,ncn=1,ccn=4 -o pascom,mdlbond4 -d wh
    input.csv output.sdf
```

---

[2]https://ftp.ncbi.nlm.nih.gov/pubchem/Bioassay/Extras/bioassays.tsv.gz , Accessed Jun 2024

Tab. 2: Node features

| Indices | Description |
|---------|-------------|
| 0-11 | One-hot encoding of element type: H, C, N, O, F, Si, P, S, Cl, Br, I, other |
| 12-15 | One-hot encoding of node degree: 1, 2, 3, 4 |
| 16 | Formal charge |
| 17 | Is in a ring |
| 18 | Is aromatic |
| 19 | Explicit valence |
| 20 | Atom mass |
| 21 | Gasteiger charge |
| 22 | Gasteiger H charge |
| 23 | Crippen contribution to logP |
| 24 | Crippen contribution to molar refractivity |
| 25 | Total polar sufrace area contribution |
| 26 | Labute approximate surface area contribution |
| 27 | EState index |

Tab. 3: Edge features

| Indices | Description |
|---------|-------------|
| 0 | Is aromatic |
| 1 | Is conjugate |
| 2 | Is in a ring |
| 3-6 | One-hot encoding of bond type: 1, 1.5, 2, 3 |

### 1.4.2 Node and Edge Attributes in the Graph Format

The 2D and 3D graphs both contain node features. Since the edge of 2D graphs are defined to be chemical bonds, those edges have chemical bond features associated. 3D graphs defines a edge existence based on 3D Euclidean distance within a distance cutoff. Therefore 3D graph has no edge features but and only has two states: outside the cutoff (no edge) or within the cutoff (has edge).

We use the node and edge features from [34], as listed in Tab. 2 and Tab. 3. However, note that these are general molecular features. We encourage the community to design more specialized features for specific tasks to better advance the drug discovery field.

### 1.5 Additional Experimental Measurement

Our data is curated from actual biochemical experiments stored and detailed on PubChem database. The floating-valued labels correspond to measured activity values, typically expressed as EC50 or IC50 in the unit of micromolar (μM). These measurements, requiring extensive biochemical experiments, are more expensive to obtain compared to binary active/inactive labels, so they are usually only available for confirmed active compounds to manage costs effectively. For other compounds, the data remains binary (active/inactive).

Of the nine datasets, three include these costly measurements for confirmed actives, providing them with floating-value labels. Active compounds in these datasets generally show activity in the tens of micromolar range (detailed below). To represent inactive compounds lacking floating-value measurements, we assign a value of 1000 μM (or 1 mM), which is commonly accepted as inactive in drug discovery.

We acknowledge that setting this artificial value of 1000 μM could skew the results. The chosen value of 1000 μM was selected as a default to indicate inactivity, but it may not be optimal for all use cases.

We encourage researchers with a deep understanding of drug discovery to leverage the floating-value information in the dataset and consider adjusting the default value of 1000 μM according to their

specific needs. For instance, they might choose to set this value to the maximum detectable activity or another relevant threshold. For other researchers, as noted in the paper, we recommend sticking to binary classification tasks, especially if they are not comfortable with the implications of using this artificial value.

We list the specifications of these three datasets with additional experimental measures.

**AID435008**: The additional experimental measure data was extracted from AID504699 and AID504701. The IC50 values were averaged from those assays (given their identical experimental condition). The final IC50 data contains 247 unique compounds (before other filters), stored in the "activity_value" in the .csv files with unit μM.

**AID2689**: The additional experimental measure data was extracted from AID2821. The final EC50 data contains 224 unique compounds(before other filters), stored in the "activity_value" in the .csv files with unit μM.

**AID488997**: The additional experimental measure data was extracted from AID588401 and AID504840. The IC50 values were averaged from those assays (given their identical experimental condition). The final IC50 data contains 896 unique compounds (before other filters), stored in the "activity_value" in the .csv files with unit μM.

## 1.6 Visualization of the Chemical Space

A example T-SNE visualization of chemical space before and after curation was created and shown in Fig. 12. We could observe two key points:

1. The number of actives is significantly reduced after curation, highlighting the high false positive rate in the primary screen. This underscores the importance of our pipeline in identifying confirmed actives.

2. After curation, the actives are well-distributed across the chemical space, rather than being clustered. This broad representation is crucial for training robust AI models.

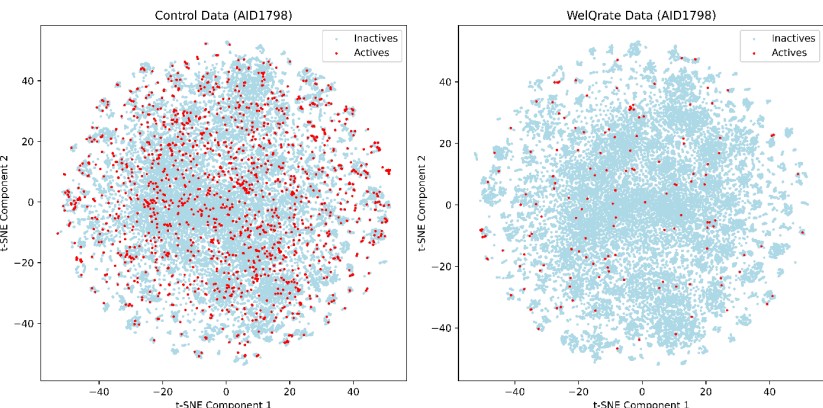

Fig. 12: A T-SNE visualization of the ECPF4 embedding of AID1798, before and after curation.

# 2  Evaluation Framework

## 2.1  Evaluation Metric Details

### Logarithmic Receiver-Operating-Characteristic Area Under the Curve with the False Positive Rate in the Range [0.001, 0.1] ($\textbf{logAUC}_{[0.001,0.1]}$)

Ranged LogAUC ([38]) features in a high decision cutoff corresponding to the left side of the Receiver-Operating-Characteristic (ROC) curve, i.e., those False Positive Rates (FPRs) with small values. Also, because the threshold cannot be predetermined, the area under the curve is used to consolidate all possible thresholds within a certain small FPR range. Finally, the logarithm is used to bias towards smaller FPRs. Following prior work ([34, 18]), we choose the FPR range between 0.001 and 0.1. A perfect classifier achieves a $\text{logAUC}_{[0.001,0.1]}$ of 1, while a random classifier reaches a $\text{logAUC}_{[0.001,0.1]}$ of around 0.0215, as shown below:

$$\frac{\int_{0.001}^{0.1} x \mathrm{d}\log_{10} x}{\int_{0.001}^{0.1} 1 \mathrm{d}\log_{10} x} = \frac{\int_{-3}^{-1} 10^u \mathrm{d}u}{\int_{-3}^{-1} 1 \mathrm{d}u} \approx 0.0215$$

### Boltzmann-Enhanced Discrimination of Receiver Operating Characteristic (BEDROC)

BEDROC ([41]) bounded by the interval [0,1], emphasizes the model's ability to rank active compounds early in the prediction list. It is derived from the robust initial enhancement ($RIE$), its minimum value $RIE_{min}$ (when all the actives are ranked at the tail of the list), and its maximum value $RIE_{max}$ (when all the actives are ranked at the beginning of the list), defined as follows:

$$RIE = \frac{\frac{1}{n}\sum_{i=1}^{n} e^{-\alpha x_i}}{\frac{1}{N}\left(\frac{1-e^{-\alpha}}{e^{\alpha/N}-1}\right)}$$

$$RIE_{max} = \frac{1 - e^{-\alpha R_a}}{R_a\left(1 - e^{-\alpha}\right)}$$

$$RIE_{min} = \frac{1 - e^{\alpha R_a}}{R_a\left(1 - e^{\alpha}\right)}$$

Where $n$ and $N$ are the numbers of actives and total compounds tested, respectively. $x_i$ is the relative rank of the $i$th active such that $x_i = r_i/N$ for $r_i$ being its rank in the prediction list. $R_\alpha$ is the ratio of actives $(n/N)$, and $\alpha$ is a tunable parameter that controls the metric's sensitivity to early recognition. we used the recommended value of $\alpha$=20 as suggested in the original paper. The BEDROC score is calculated as:

$$BEDROC = \frac{RIE - RIE_{min}}{RIE_{max} - RIE_{min}} = \frac{\sum_{i=1}^{n} -e^{r_i/N}}{\frac{n}{N}\left(\frac{1-e^{-\alpha}}{e^{\alpha/N}-1}\right)} \times \frac{R_a \sinh\left(\alpha/2\right)}{\cosh\left(\alpha/2\right) - \cosh\left(\alpha/2 - \alpha R_a\right)} + \frac{1}{1 - e^{\alpha(1-R_a)}}$$

### Enrichment Factor with Cutoff 100 ($\textbf{EF}_{100}$)

Enrichment factor ([22]) is often used metric in virtual screening. It measures how well a screening method can increase the proportion of active compounds in a selection set, compared to a random selection set. Here we select the top 100 compounds as the selection set. And the $EF_{100}$ can be defined as follows:

$$EF_{100} = \frac{n_{100}/N_{100}}{n/N}$$

where $n_{100}$ is the number of true active compounds in the ranked top 100 predicted compounds given by the model, $N_{100}$ is the number of compounds in the top 100 predicted compounds (i.e., 100), $n$ is the number of active compounds in the entire dataset, $N$ is the number of compounds in the entire dataset. It is essentially a measure of the model's ability to "enrich" the set of compounds for further testing. A random selection set receives an $EF_{100}$ of 1. If no true active compounds are in the top 100 compounds, the $EF_{100}$ becomes 0.

### Discounted Cumulative Gain with Cutoff 100 ($\textbf{DCG}_{100}$)

DCG ([27]) is a measure of ranking quality often used in web search. In a web search, it is obvious that a method is better when it positions highly relevant documents at the top of the search results. Virtual screening has a similar evaluation logic where we desire the active molecules to appear at the top of the selection set. To calculate DCG, a simpler version metric named Cumulative Gain (CG) ([27]) is introduced below. CG is the sum of the relevance value of a compound in the selection set. In our case, a true active compound receives a

relevance value of 1, while a true inactive compound receives a relevance value of 0. So, the CG with cutoff 100 ($CG_{100}$) equals the number of true active compounds in the top 100 compounds, i.e.,

$$CG_{100} = \sum_{i=1}^{100} y_i$$

It can be observed that $CG_{100}$ is unaffected by changes in the ordering of compounds. DCG hence aims to penalize a true active molecule appearing lower in the selection set by logarithmically reducing the relevance value proportional to the predicted rank of the compound, i.e.,

$$DCG_{100} = \sum_{i=1}^{100} y_i/log_2(i+1)$$

## 2.2 Cross-Validation

Traditional cross-validation is a technique used to assess the performance of a machine learning model by partitioning the dataset into a number of equally sized subsets or *folds*. The process typically involves dividing the entire dataset into $k$ folds, where one fold is set aside as the testing set and the remaining $k-1$ folds form the training set. This process is repeated $k$ times, with each fold used exactly once as the testing set. The performance metric is calculated for each of the $k$ iterations and then averaged to produce a single estimation of the model's performance. Traditional cross-validation allows the model to be trained and tested on different subsets of the data, providing a more robust estimate of the model's performance compared to a single train-test split. This approach helps reduce overfitting and offers insights into how the model generalizes to independent datasets. This is particularly important for *WelQrate* dataset collection because the number of active compounds in reality is limited, and evaluating the performance on just one test set would be biased.

Nested cross-validation is an advanced technique used to evaluate the performance of machine learning models, especially when hyperparameter tuning is involved. It consists of two levels of cross-validation: an outer loop for model evaluation and an inner loop for hyperparameter tuning. The outer loop divides the dataset into $k$ folds, with each fold serving as the test set while the remaining folds form the training set. Within each outer loop iteration, the inner loop further splits the training set into $m$ folds to perform hyperparameter tuning, ensuring the test data is never used in the tuning process. The optimal hyperparameters from the inner loop are used to train the model on the entire outer training set, which is then evaluated on the outer test set. This process is repeated $k$ times, and the performance metrics from all iterations are averaged to provide a final performance estimate. Nested cross-validation provides an unbiased and reliable assessment of model performance by preventing data leakage and overfitting, making it particularly useful for selecting the best model and hyperparameters.

However, nested cross-validation is computationally expensive due to the many loops it involves. Therefore, we employ an adapted cross-validation approach, where the validation set is always fixed as the fold immediately prior to the test set (in cases where the test set is the first fold, the validation set becomes the last fold). This method strikes a balance between computational efficiency and rigorous evaluation.

# 3 Experimental Benchmarking

## 3.1 Model Details

### 3.1.1 Descriptor Used in Domain Baseline Model

The descriptor used in the domain baseline is generated with BCL. There are three types of features in the generated descriptor: scalar, signed 2D autocorrelations (2DA_Sign) [49], and signed 3D autocorrelations (3DA_Sign) [49].

The original unsigned version of 2D_Sign is proposed in [37] (denoted as 2DA). The original unsigned version of 3D_Sign is proposed in [9] (denoted as 3DA). 2DA defines bond distance as the distance measure, while the 3DA defines the 3D Euclidean distance as the distance measure. The 2DA and 3DA calculate the autocorrelation value for a distance bin between $r_a$ and $r_b$ based on the following formula:

$$Autocorrelation(r_a, r_b) = \sum_i^n \sum_j^n \mathbb{1}(r_{ij}) P_i P_j \tag{1}$$

where $i$ and $j$ are two nodes (atoms), $n$ is the total number of nodes, $P$ is an atomic property, $\mathbb{1}$ is an indicator function, which evaluates to 1 if the distance $r_{ij}$ satisfies $r_a \leq r_{ij} < r_b$, otherwise evaluates to 0. 2DA_Sign and 3DA_Sign extend the original 2DA and 3DA by having 3 values for each distance bin, corresponding to the positive-positive, positive-negative, and negative-negative pairs of atomic properties, to circumvent the situation where a negative and a positive cancels out during the summation. The atomic properties used in 2DA_Sign and 3DA_Sign are listed in Tab.4.

There are 23 scalars (see Tab. 4), each taking one dimension.

The 2DA_Sign evaluates up to 11 bonds (exclusive). It has 32 one-dimensional values corresponding to 11 distance bins (3 values per bin originally take 33 dimensions. However, the first bin is for 0 bonds away, essentially the square of the property of an atom, which has no positive-negative pairs, therefore excluded) for each atomic property. With 4 atomic properties, there are 32*4=128 dimensions for 2DA_Sign.

The 3DA_Sign evaluates up to 6 Å(exclusive). There are 24 distance bins for a step size of 0.25 Å. However, the distance bin [0, 0.25) reduces to 2D_Sign, therefore excluded. [0.25, 0.5), [0.5, 0.75], [0.75, 1) are generally evaluates to 0, therefore are excluded as well. This results in a total number of 20 distance bins. Each bin has 3 values so there are 60 one-dimensional value for each atomic property. With 4 atomic properties, there are 60*4=240 dimensions for 3DA_Sign.

The final descriptor is of 391 dimensions (23+128+240 = 391).

Tab. 4: Details of the descriptor used in the domain baseline.

| Feature | Type |
|---|---|
| Molecular Weight | |
| Hydrogen Bond Donor | |
| Hydrogen Bond Acceptor | |
| LogP | |
| Total Charge | |
| Number of Rotatable Bonds | |
| Number of Aromatic Rings | |
| Number of Rings | |
| Topological Polar Surface Area | |
| Girth - Widest Diameter of Molecule (Å) | |
| Bond Girth - Maximum Number of Bonds Between Two Atoms | |
| Number of Atoms in the Largest Ring | Scalar |
| Number of Atoms in the Smallest Ring | |
| Number of Atoms in Aromatic Fused Ring | |
| Number of Atoms in Fused Ring | |
| Min of Sigma Charge | |
| Max of Sigma Charge | |
| Standard Deviation of Sigma Charge | |
| Sum of Absolute Values of Sigma Charge | |
| Min of V Charge | |
| Max of V Charge | |
| Standard Deviation of V Charge | |
| Sum of Absolute Values of V Charge | |
| Atomic Sigma Charge | |
| Atomic VCharge | |
| Is H (1 for H, -1 for Heavy Atoms) | 2DA_Sign |
| Is In Aromatic Ring (1 for Yes, -1 for No) | |
| Atomic Sigma Charge | |
| Atomic VCharge | |
| Is H (1 for H, -1 for Heavy Atoms) | 3DA_Sign |
| Is In Aromatic Ring (1 for Yes, -1 for No) | |

## 3.2 Hyperparameters

Initially, all models undergo fine-tuning on either random split 1 or scaffold split seed 1 to determine optimal hyperparameters for each dataset. These identified hyperparameters are then applied to training on other data splits across various experiments. The following are the hyperparameter pools for all models, all datasets across two splits. All models were trained using the AdamW optimizer with a default weight decay ratio of 0.01. Additionally, we implemented a learning rate scheduler featuring a polynomial decay with a linear warmup phase. The learning rate linearly increases from 0 to the peak learning rate during the 2000 warmup iterations, then decays polynomially to the end learning rate $10^{-9}$ over the total number of iterations, following a power law. For SchNet, DimeNet++, and SphereNet, we referred to the search space reported by [33]. We applied an early stopping limit of 30 epochs to halt training if the validation logAUC does not improve for 30 consecutive epochs. [33] and the default hyperparameters in the *torch-geometric* [17] and *DGL* [53] libraries. Here, we present the tunable hyperparameter pools for all models, and in each of the following tables.

Next, we present the tuned hyperparameters for each model under random and scaffold split across all datasets. The tuned hyperparameters for each model are represented as a list, where each value corresponds to the hyperparameters in the pool with **the same order**.

Tab. 5: Hyperparameter pool for the Naive Model.

| Hyperparameters | Search Space & Values |
| --- | --- |
| Peak Learning Rate | $0.001, 0.0001, 0.00001$ |
| Hidden Channels | $32, 64, 128$ |
| Number of Layers | $3, 4, 5$ |

Tab. 6: Hyperparameter pool for the Domain Model.

| Hyperparameters | Search Space & Values |
| --- | --- |
| Peak Learning Rate | $0.001, 0.0001, 0.00001$ |
| Hidden Channels | $32, 64, 128$ |
| Number of Layers | $3, 4, 5$ |

Tab. 7: Hyperparameter pool for the GCN.

| Hyperparameters | Search Space & Values |
| --- | --- |
| Peak Learning Rate | $0.01, 0.001, 0.0001$ |
| Number of Layers | $2, 3, 4, 5$ |
| Hidden Channels | $16, 32, 64, 128$ |

Tab. 8: Hyperparameter pool for the GIN.

| Hyperparameters | Search Space & Values |
| --- | --- |
| Peak Learning Rate | $0.01, 0.001, 0.0001$ |
| Number of Layers | $2, 3, 4, 5$ |
| Hidden Channels | $16, 32, 64, 128$ |

Tab. 9: Hyperparameter pool for the GAT.

| Hyperparameters | Search Space & Values |
| --- | --- |
| Peak Learning Rate | $0.01, 0.001, 0.0001$ |
| Number of Layers | $3, 4, 5$ |
| Heads | $4, 8, 16$ |
| Hidden Channels | $16, 32, 64, 128$ |

Tab. 10: Hyperparameter pool for the SchNet.

| Hyperparameters | Search Space & Values |
| --- | --- |
| Peak Learning Rate | $0.0001, 0.001, 0.01$ |
| Number of Layers | $5, 6, 7$ |
| Hidden Channels | $32, 64$ |
| Number of Filters | $16, 32, 64$ |
| Number of Gaussians | $25, 50, 75$ |

Tab. 11: Hyperparameter pool for the SMILES2Vec.

| Hyperparameters | Search Space & Values |
| --- | --- |
| Peak Learning Rate | $0.001, 0.0001$ |
| Embedding Dimension | $25, 50, 100$ |
| Kernel Size | $3, 5, 7$ |
| Number of Strides | $1, 2, 3$ |

Tab. 12: Hyperparameter pool for the TextCNN.

| Hyperparameters | Search Space & Values |
| --- | --- |
| Peak Learning Rate | $0.001, 0.0001$ |
| Embedding Dimension | $25, 50, 75, 100$ |

Tab. 13: Hyperparameter pool for the DimeNet++ model.

| Hyperparameters | Search Space & Values |
| --- | --- |
| Peak Learning Rate | $0.001, 0.0005, 0.0001$ |
| Number of Blocks | $4, 5, 6$ |
| Basis Embedding Size | $6, 8$ |
| Number of Spherical Harmonics | $3, 5, 7$ |
| Number of Radial Basis Functions | $4, 6, 8$ |

Tab. 14: Hyperparameter pool for the SphereNet model.

| Hyperparameters | Search Space & Values |
|---|---|
| Peak Learning Rate | $0.001, 0.0001$ |
| Number of Layers | $2, 3, 4$ |
| Basis Embedding Size (Distance) | $4, 8$ |
| Basis Embedding Size (Angle) | $4, 8$ |
| Basis Embedding Size (Torsion) | $4, 8$ |
| Number of Spherical Harmonics | $3, 5$ |

| GCN | CV Hyperparameter List | Scaffold Hyperparameter List |
|---|---|---|
| AID1798 | [0.001, 128, 5] | [0.01, 16, 5] |
| AID1843 | [0.0001, 64, 3] | [0.0001, 32, 3] |
| AID2258 | [0.001, 128, 3] | [0.01, 64, 3] |
| AID2689 | [0.001, 128, 4] | [0.01, 64, 2] |
| AID435008 | [0.01, 32, 3] | [0.01, 128, 4] |
| AID435034 | [0.01, 32, 3] | [0.001, 64, 4] |
| AID463087 | [0.01, 128, 2] | [0.001, 128, 4] |
| AID485290 | [0.001, 64, 3] | [0.01, 128, 3] |
| AID488997 | [0.001, 128, 4] | [0.001, 128, 5] |

| GIN | CV Hyperparameter List | Scaffold Hyperparameter List |
|---|---|---|
| AID1798 | [0.001, 64, 4] | [0.0001, 32, 3] |
| AID1843 | [0.0001, 128, 2] | [0.0001, 32, 2] |
| AID2258 | [0.001, 128, 3] | [0.001, 128, 3] |
| AID2689 | [0.0001, 64, 4] | [0.0001, 128, 4] |
| AID435008 | [0.01, 64, 4] | [0.01, 32, 2] |
| AID435034 | [0.001, 64, 2] | [0.001, 64, 3] |
| AID463087 | [0.001, 128, 4] | [0.001, 128, 3] |
| AID485290 | [0.001, 32, 4] | [0.001, 32, 3] |
| AID488997 | [0.001, 64, 4] | [0.01, 32, 4] |

| GAT | CV Hyperparameter List | Scaffold Hyperparameter List |
|---|---|---|
| AID1798 | [0.01, 5, 8, 32] | [0.01, 5, 4, 128] |
| AID1843 | [0.01, 5, 8, 64] | [0.0001, 4, 4, 128] |
| AID2258 | [0.001, 5, 8, 128] | [0.01, 5, 8, 64] |
| AID2689 | [0.001, 5, 8, 32] | [0.001, 4, 4, 128] |
| AID435008 | [0.01, 4, 8, 64] | [0.01, 5, 4, 128] |
| AID435034 | [0.01, 5, 4, 32] | [0.001, 5, 8, 128] |
| AID463087 | [0.001, 4, 8, 128] | [0.001, 5, 4, 64] |
| AID485290 | [0.01, 5, 8, 128] | [0.01, 5, 8, 128] |
| AID488997 | [0.01, 3, 4, 128] | [0.001, 4, 8, 128] |

| Naive Model | CV Hyperparameter List | Scaffold Hyperparameter List |
|---|---|---|
| AID1798 | [0.001, 64, 4] | [0.0001, 32, 3] |
| AID1843 | [0.0001, 128, 5] | [0.001, 64, 3] |
| AID2258 | [0.001, 128, 4] | [0.001, 128, 5] |
| AID2689 | [0.001, 128, 3] | [0.0001, 32, 5] |
| AID435008 | [0.001, 64, 5] | [0.0001, 32, 3] |
| AID435034 | [0.0001, 128, 3] | [1e-05, 32, 5] |
| AID463087 | [0.001, 128, 5] | [0.001, 128, 4] |
| AID485290 | [0.001, 32, 4] | [0.001, 128, 3] |
| AID488997 | [0.001, 128, 3] | [0.001, 128, 3] |

| Domain Model | CV Hyperparameter List | Scaffold Hyperparameter List |
| --- | --- | --- |
| AID1798 | [0.001, 128, 5] | [0.001, 128, 4] |
| AID1843 | [0.0001, 64, 3] | [0.0001, 32, 5] |
| AID2258 | [0.001, 128, 3] | [0.001, 32, 3] |
| AID2689 | [0.0001, 128, 3] | [0.001, 64, 4] |
| AID435008 | [0.0001, 32, 3] | [0.001, 64, 3] |
| AID435034 | [0.0001, 128, 3] | [1e-05, 64, 3] |
| AID463087 | [0.001, 128, 3] | [0.001, 128, 3] |
| AID485290 | [0.001, 128, 3] | [0.001, 32, 3] |
| AID488997 | [0.0001, 128, 4] | [0.0001, 64, 4] |

| SMILES2Vec | CV Hyperparameter List | Scaffold Hyperparameter List |
| --- | --- | --- |
| AID1798 | [0.001, 50, 3, 2] | [0.001, 50, 5, 1] |
| AID1843 | [0.001, 100, 7, 2] | [0.001, 50, 5, 3] |
| AID2258 | [0.001, 100, 5, 2] | [0.001, 25, 5, 2] |
| AID2689 | [0.001, 50, 5, 3] | [0.001, 25, 7, 3] |
| AID435008 | [0.001, 50, 3, 2] | [0.001, 25, 7, 3] |
| AID435034 | [0.0001, 50, 5, 2] | [0.001, 50, 7, 3] |
| AID463087 | [0.0001, 100, 7, 1] | [0.001, 50, 5, 1] |
| AID485290 | [0.001, 25, 3, 3] | [0.001, 50, 5, 1] |
| AID488997 | [0.0001, 50, 3, 2] | [0.0001, 50, 7, 3] |

| TextCNN | CV Hyperparameter List | Scaffold Hyperparameter List |
| --- | --- | --- |
| AID1798 | [0.0001, 75] | [0.001, 50] |
| AID1843 | [0.0001, 50] | [0.001, 75] |
| AID2258 | [0.0001, 75] | [0.001, 75] |
| AID2689 | [0.001, 50] | [0.0001, 100] |
| AID435008 | [0.0001, 75] | [0.001, 25] |
| AID435034 | [0.001, 100] | [0.001, 100] |
| AID463087 | [0.0001, 100] | [0.001, 100] |
| AID485290 | [0.0001, 75] | [0.0001, 100] |
| AID488997 | [0.001, 100] | [0.001, 100] |

| SchNet | CV Hyperparameter List | Scaffold Hyperparameter List |
| --- | --- | --- |
| AID1798 | [0.001, 5, 64, 16, 75] | [0.001, 7, 64, 16, 50] |
| AID1843 | [0.001, 6, 64, 16, 25] | [0.001, 7, 32, 32, 75] |
| AID2258 | [0.001, 6, 64, 16, 50] | [0.001, 5, 64, 16, 75] |
| AID2689 | [0.001, 6, 32, 16, 75] | [0.001, 6, 32, 64, 25] |
| AID435008 | [0.001, 5, 64, 32, 50] | [0.001, 6, 64, 64, 50] |
| AID435034 | [0.001, 5, 32, 32, 50] | [0.001, 6, 32, 64, 75] |
| AID463087 | [0.001, 5, 64, 64, 25] | [0.001, 5, 32, 64, 25] |
| AID485290 | [0.001, 6, 32, 32, 25] | [0.001, 6, 32, 16, 75] |
| AID488997 | [0.001, 5, 64, 16, 25] | [0.001, 7, 32, 64, 25] |

| DimeNet++ | CV Hyperparameter List | Scaffold Hyperparameter List |
| --- | --- | --- |
| AID1798 | [0.0001, 6, 6, 3, 6] | [0.0005, 4, 8, 7, 8] |
| AID1843 | [0.0001, 6, 6, 3, 6] | [0.0001, 4, 6, 3, 8] |
| AID2258 | [0.0001, 4, 6, 5, 4] | [0.001, 4, 6, 5, 4] |
| AID2689 | [0.0001, 5, 6, 5, 6] | [0.0001, 5, 8, 5, 8] |
| AID435008 | [0.0001, 4, 6, 3, 4] | [0.001, 5, 6, 5, 4] |
| AID435034 | [0.0001, 6, 6, 5, 4] | [0.0001, 5, 6, 3, 4] |
| AID463087 | [0.0001, 4, 6, 3, 4] | [0.0001, 6, 8, 3, 4] |
| AID485290 | [0.001, 6, 8, 3, 6] | [0.0001, 6, 6, 5, 8] |
| AID488997 | [0.0001, 5, 6, 3, 4] | [0.001, 4, 8, 3, 4] |

| **SphereNet** | CV Hyperparameter List | Scaffold Hyperparameter List |
|---|---|---|
| AID1798 | [0.001, 1, 4, 4, 4, 3] | [0.0001, 3, 8, 8, 4, 5] |
| AID1843 | [0.001, 4, 4, 4, 4, 3] | [0.001, 4, 8, 8, 4, 3] |
| AID2258 | [0.0001, 4, 4, 4, 4, 3] | [0.001, 4, 8, 4, 4, 3] |
| AID2689 | [0.001, 4, 8, 8, 4, 3] | [0.001, 4, 4, 4, 4, 3] |
| AID435008 | [0.001, 4, 8, 8, 4, 5] | [0.001, 4, 8, 8, 4, 3] |
| AID435034 | [0.001, 4, 4, 8, 4, 5] | [0.001, 4, 8, 8, 4, 5] |
| AID463087 | [0.001, 4, 4, 4, 4, 3] | [0.001, 4, 8, 8, 4, 5] |
| AID485290 | [0.001, 4, 4, 8, 4, 5] | [0.001, 4, 4, 4, 4, 3] |
| AID488997 | [0.001, 4, 4, 8, 4, 3] | [0.001, 3, 8, 4, 4, 5] |

### 3.3 Training Details

In our current work, we addressed class imbalance by employing a straightforward oversampling approach. Specifically, during model training, we used a strategy where molecules were sampled based on a probability inversely proportional to their class label frequency. This ensured that in each training batch, there was a roughly equal number of molecules from both classes, thereby mitigating the imbalance.

We recognize that class imbalance is a complex problem, and while oversampling is a practical approach, it is not the only solution. In fact, our previous research has explored dedicated methods for handling imbalance in graph-based models, including imbalanced graph classification. We agree that this is an interesting area for further exploration, and we will include a discussion in the paper about potential future directions beyond oversampling, such as the use of advanced techniques like undersampling, synthetic data generation, and specialized algorithms for imbalanced data.

# 4 Additional Contents

## 4.1 Assets and Computing Resources

### 4.1.1 Asset Licenses

PubChem is the source of data in this work and it is a free to use database[3].

BCL, which is used to generate the domain-driven descriptor, is under MIT license[4].

RDkit, which is used for most part of the data processing, is under the 3-Clause BSD License[5].

Corina v5.0, that is used to generated SDF, is a commercial software and is licensed to the Meiler lab.

All of our experiments are executed with Python. Python is developed under an OSI-approved open-source license, making it freely usable and distributable, even for commercial use. Python's license is administered by the Python Software Foundation[6].

Graph formats are generated with PyTorch Geometric under the MIT license[7].

The dataset diagrams in the main paper and supplement were created with BioRender.com, with an academic license.

### 4.1.2 Computing Resources

The experiments were carried out on a DGX cluster equipped with an AMD EPYC 7742 64-Core CPU processor, 2.0TiB of RAM, and eight NVIDIA A100-SXM4 GPUs, each with 80GB of memory.

---

[3]https://pubchem.ncbi.nlm.nih.gov/docs/downloads, Accessed Jun 2024

[4]https://github.com/BCLCommons/bcl/blob/master/LICENSE, Accessed Jun 2024

[5]https://github.com/rdkit/rdkit/blob/master/license.txt, Accessed Jun 2024

[6]https://www.python.org/about/, Accessed Jun 2024

[7]https://github.com/pyg-team/pytorch_geometric/blob/master/LICENSE, Accessed Jun 2024