# OpenReview forum: "WelQrate: Defining the Gold Standard in Small Molecule Drug Discovery Benchmarking"
_NeurIPS.cc/2024/Datasets_and_Benchmarks_Track — NeurIPS 2024 Track Datasets and Benchmarks Poster_

### Official Review · Reviewer_E61p · 2024-07-14
**Good paper, accept**

**Rating:** 8
**Confidence:** 4
**Correctness:** Yes
**Clarity:** Yes

**Review:**

Quality: This work is of high quality, developing a robust evaluation framework. The attention in data curation and the comprehensive benchmarking of existing models underscore the thoroughness of the research.

Clarity: The paper is clear and well-organized, making it easy to understand the contributions and significance of WelQrate. The description of the data curation pipeline, evaluation framework, and benchmarking process is concise and informative.

Originality: The introduction of WelQrate as a new gold standard for benchmarking in small molecule drug discovery is highly original. The authors address a critical gap in the current AI drug discovery landscape by providing a realistic and comprehensive evaluation framework.

pros:
1. A significant strength of this paper is that the dataset covers highly imbalanced activity labels, reflecting the low hit rates typically encountered in real-world drug discovery.
2. A rigorous curation pipeline to ensure the data quality which is crucial for reliable benchmarking.

cons:
Some points included in the opportunities for improvement section

**Strengths:**

These improvements could significantly benefit the AI in chemistry community by providing more robust, diverse, and practical benchmarks, ultimately advancing the development and application of AI models in drug discovery.

**Additional Feedback:**

Null

**Documentation:**

Yes

**Ethics:**

No concerns

**Limitations:**

Yes

**Opportunities For Improvement:**

1. A first suggestion is visulazing the impact of data curation workflow on data itself. While the performance differences might not always be obvious due to varying molecule coverage, it is highly recommended to include discussions and experiments on data quality from the perspective of data itself. This can be achieved by visualizing the cleaned and uncleaned data space, graph representation to highlight this necessity.

2. Given the use of 3D coordinates, I highly recommend comparing different methods to Corina v5.0 used in this work. Additionally, since calculations can vary in accuracy, it may be necessary to compare these 3D structures across different calculation levels to assess their reliability.

**Relation To Prior Work:**

Yes

**Summary And Contributions:**

The paper introduces WelQrate, a new standard for benchmarking in small molecule drug discovery, addressing the need for better model evaluation frameworks. It contributes a curated dataset collection, a standardized evaluation framework and benchmarks existing deep learning architectures, demonstrating the importance of high-quality labeling. WelQrate provides an important scope for robust and standardized evaluation.

---

> ### Author Rebuttal · Authors · 2024-08-17
>
> Thank you for these thoughtful suggestions. The responses to your questions can be seen below.
>
> **1. A first suggestion is visulazing the impact of data curation workflow on data itself. While the performance differences might not always be obvious due to varying molecule coverage, it is highly recommended to include discussions and experiments on data quality from the perspective of data itself. This can be achieved by visualizing the cleaned and uncleaned data space, graph representation to highlight this necessity.**
>
> Thank you for the great suggestion. A T-SNE visualization of chemical space before and after curation was created and shown in the attached PDF. We could observe two key points:
>
> 1. The number of actives is significantly reduced after curation, highlighting the high false positive rate in the primary screen. This underscores the importance of our pipeline in identifying confirmed actives.
>
> 2. After curation, the actives are well-distributed across the chemical space, rather than being clustered. This broad representation is crucial for training robust AI models.
>
> We appreciate your suggestion and will incorporate a discussion on these observations in the final manuscript.
>
> ---
>
> **2. Given the use of 3D coordinates, I highly recommend comparing different methods to Corina v5.0 used in this work. Additionally, since calculations can vary in accuracy, it may be necessary to compare these 3D structures across different calculation levels to assess their reliability.**
>
> Thank you for the suggestion. We plan to address this in future work, including comparing conformation generation methods, such as ETKDG [1] in RDKit and deep learning-based approaches like torsional diffusion [2]. Currently while most current studies leave 3D conformation generation to ML practitioners, our goal is to provide a standard 3D conformation, assuming the molecule is in a solution at its lowest energy state. We plan to explore beyond standard 3D conformations in future WelQrate versions, using tools like BCL::conf [3], which generates likely 3D binding conformations based on ligand statistics from the Crystallography Open Database [4] to further help benchmarking practical models to advance the drug discovery field.
>
> [1] Riniker, Sereina, and Gregory A. Landrum. "Better informed distance geometry: using what we know to improve conformation generation." Journal of chemical information and modeling 55.12 (2015): 2562-2574.
>
> [2] Jing, Bowen, et al. "Torsional diffusion for molecular conformer generation." Advances in Neural Information Processing Systems 35 (2022): 24240-24253.
>
> [3] Mendenhall, Jeffrey, et al. "BCL:: Conf: improved open-source knowledge-based conformation sampling using the crystallography open database." Journal of chemical information and modeling 61.1 (2020): 189-201.
>
> [4] Gražulis, Saulius, et al. "Crystallography Open Database–an open-access collection of crystal structures." Journal of applied crystallography 42.4 (2009): 726-729.

---

> > ### Comment · Reviewer_E61p · 2024-08-19
> >
> > Thank you for incorporating my feedback, and for the clarifications.

---

> > > ### Author Response · Authors · 2024-08-26
> > >
> > > Thank you for your thoughtful feedback and for reviewing our clarifications. We're glad to have addressed your concerns and appreciate your contributions to improving our work.

---

### Official Review · Reviewer_yZji · 2024-07-19
**The work proposes a hierarchical curation method for generating datasets of importance to therapeutric compounds**

**Rating:** 4
**Confidence:** 4
**Clarity:** The writing lacks clarity and could b…

**Review:**

Distributed across other boxes under strengths/weakness etc.

**Strengths:**

a. Introduction of WelQrate as a new standard for benchmarking in small molecule drug discovery.
b. Rigorous data curation process.
c. Comprehensive evaluation framework covering featurization, 3D structure generation, and relevant metrics for accurate and realistic virtual screening evaluations.
d. Clear explanation of the 3D graph generation process

**Additional Feedback:**

Please see above

**Correctness:**

Mostly correct, but limitations makes me remain less enthusiastic in commenting here.

**Documentation:**

Yes, the dataset is available on a public website.

**Ethics:**

No issues

**Limitations:**

a. Therapeutics Data Commons (TDC) offers 66 AI/ML-ready datasets, whereas WelQrate includes only 9 datasets. Your data curation pipeline employs hierarchical curation and various filters to ensure high-quality data. It would be advantageous to apply the same filters used in WelQrate to the 66 TDC datasets, as this could uncover many important datasets. Is this worth considering?
b. In Figure 4, the authors present the results of the 2D-graph molecular representation using three different models: Graph Convolutional Neural Network (GCN), Graph Isomorphism Network (GIN), and Graph Attention Network (GAT). How do the overall performance metrics (logAUC/BEDROC) reflect the individual performance of these three models? Did the authors use mean values or another method?
c. The statement '2D and 3D Graph-Based models trained on the control dataset actually outperform in logAUC and BEDROC' is somewhat fuzzy. The control dataset contains some noisy data, such as 'invalid chemical structures, inconsistent chemical representations, undefined stereochemistry, and poorly defined endpoints'. How can noisy data lead to better performance? The performance comparison seems inappropriate. It would be better to test all 9 datasets to draw stronger conclusions.
d. The authors hypothesize that 'this could be related to an overfitting issue in these complex models'. Why do the authors assume overfitting? Reliable ML practitioners should always check the performance difference between training and testing data to appeal robust conclusions.
e. The authors mention that 'predicting the activity of molecules with unseen scaffolds represents a distribution-shift problem'. Did the authors conduct any control experiments to assess distribution shifts? Have they quantified this shift?

**Opportunities For Improvement:**

a. What are the different types of 'inconsistent chemical representations' and 'undefined stereochemistry'? Can you provide specific examples to illustrate these limitations?
b. Mistake in the caption: 'Figure 4: Performance Comparison among different models (RQ1) trained respectively with WelQrate and Control Dataset (RQ2)'. In 'Comparison', 'C' should be in lowercase.
c. The use of 'Fig. 3' in the text and 'Figure 3' in the caption is inconsistent.
d. Is there a specific reason for building the ML model only for dataset AID1798?
e. Is it really the ‘Gold Standard’? Justify this one more clearly.

**Relation To Prior Work:**

Yes, it is discussed properly.

**Summary And Contributions:**

The paper presents a carefully curated collection of 9 datasets across 5 therapeutic target classes. Utilizing a hierarchical curation pipeline designed by drug discovery experts, the process includes confirmatory and counter screens, along with rigorous preprocessing steps such as PAINS filtering, to ensure accurate labeling of active molecules. The findings are compiled on the publicly accessible website WelQrate.org. The paper also benchmarks existing deep learning architectures (e.g., 2D/3D graph neural networks) on WelQrate.

---

> ### Author Rebuttal · Authors · 2024-08-17
>
> Thank you for your valuable feedback. We address the specific questions below and will revise accordingly in the final version of the paper.
>
> **1. What are the different types of 'inconsistent chemical representations' and 'undefined stereochemistry'? Can you provide specific examples to illustrate these limitations?**
>
> Thank you for your question. Inconsistent chemical representations and undefined stereochemistry are significant issues that can lead to incorrect or misleading conclusions in dataset evaluations.
>
> 1. Inconsistent Chemical Representations:
> One example of inconsistency can be found in MoleculeNet's BBBP dataset. As illustrated in Figure 1 of the attached PDF, there's a case involving the drug aspirin. In this dataset, aspirin is reported in its kekulized form, while acetylsalicylate (which is actually the same compound) is reported in its aromatic form. Despite being chemically identical, these two representations are treated as different compounds in the dataset and, ironically, they are assigned different activity labels. Such inconsistencies can lead to confusion and errors during model training and evaluation.
>
> 2. Undefined Stereochemistry:
> Undefined stereochemistry is another limitation that can significantly impact the accuracy of molecular predictions. For instance, in Figure 2 of the PDF, we provide an example where the stereochemistry of a molecule is not fully defined. This lack of definition can lead to ambiguity, as different stereoisomers of the same molecule can have vastly different biological activities. If stereochemistry is not explicitly defined, the model may incorrectly assume that different stereoisomers are the same compound, potentially leading to erroneous predictions.
>
> These examples highlight the importance of rigorous data curation and the need for careful consideration of chemical representation and stereochemistry in creating reliable datasets.
>
> ---
>
> **2. Mistake in the caption: 'Figure 4: Performance Comparison among different models (RQ1) trained respectively with WelQrate and Control Dataset (RQ2)'. In 'Comparison', 'C' should be in lowercase.**
>
> Thank you for the observation and we will correct this.
>
> ---
>
> **3. The use of 'Fig. 3' in the text and 'Figure 3' in the caption is inconsistent.**
>
> Thank you for pointing it out and we will keep consistent.
>
>
> ---
>
> **4. Is there a specific reason for building the ML model only for dataset AID1798?**
>
> For the analysis in the paper we initially selected using AID1798 as it was the smallest dataset (in terms of the number of molecules/training samples). However, based on your feedback, we have since run the same analysis across each of the datasets and these results will be included in the supplementary materials.
>
>
>
> ---
>
> **5. Is it really the ‘Gold Standard’? Justify this one more clearly.**
>
> Gold standards are typically established through widespread adoption and consensus within a community. An example is the use of Randomized Controlled Trials (RCTs) in medicine, which is considered the gold standard for determining the efficacy of a drug. We have made WelQrate widely available for adoption, have provided a means for maintaining it, and propose it as a gold standard for the community based on the rigorous, state-of-the art curation, evaluation, and benchmarking described in the paper.
>
> *Rigorous Dataset Curation:* Our curation process is exceptionally rigorous and supervised by domain experts. We have provided examples in the related works section that highlight issues in other datasets. By contrast, WelQrate ensures that the data is meticulously curated to meet strict quality standards, providing a reliable foundation for training and evaluating machine learning models.
>
> *Evaluation Framework:* We offer a unified evaluation framework that addresses key challenges in benchmarking. For instance, while most datasets provide molecular data as text sequences (e.g., SMILES), generating 3D shapes (conformations) from these sequences can vary significantly. This variability can make it difficult to compare different methods if researchers are required to generate their own 3D conformations. To mitigate this, we provide pre-generated 3D conformations, ensuring consistency across different studies. Similarly, we offer a unified featurization scheme, allowing researchers to focus on comparing model architectures without the variability introduced by different featurization methods. This approach establishes a common ground for benchmarking and promotes fair comparisons across models.
>
> *Tailored Metrics:* The metrics we propose for benchmarking are specifically designed for the unique demands of drug discovery. In this field, the cost of synthesizing and testing molecules is substantial, so the focus is on selecting the top-ranked molecules for further investigation. Our metrics, therefore, emphasize the importance of ranking performance of top-ranked molecules, akin to how search engines evaluate models according to prioritizing ranking performance of the most relevant results. This ensures that the models are evaluated based on their ability to identify the most promising candidates, which is crucial for practical applications in drug discovery.
>
> These aspects collectively make WelQrate a robust, reliable, and practical standard for benchmarking in the field of small molecule drug discovery. We believe that by addressing the key challenges in data quality, evaluation consistency, and application-specific metrics, WelQrate sets a high standard for future research in this area.

---

> > ### Author Response · Authors · 2024-08-17
> >
> > **6. Therapeutics Data Commons (TDC) offers 66 AI/ML-ready datasets, whereas WelQrate includes only 9 datasets. Your data curation pipeline employs hierarchical curation and various filters to ensure high-quality data. It would be advantageous to apply the same filters used in WelQrate to the 66 TDC datasets, as this could uncover many important datasets. Is this worth considering?**
> >
> > Thank you for the suggestion. Our focus with WelQrate is specifically on the virtual screening task, which involves predicting molecular activity based on structure. While TDC offers a broader range of tasks, such as predicting drug absorption, distribution, metabolism, excretion, and toxicity, only a few TDC datasets focus on activity prediction, and those datasets typically contain a limited number of molecules.
> >
> > Regarding the application of our curation pipeline to TDC datasets, it is not feasible for several reasons. Firstly, the tasks differ significantly; our pipeline is tailored for virtual screening tasks, while TDC covers a variety of other drug discovery tasks. Secondly, our data source is the PubChem database, which provides detailed experimental information for each entry. This allows us to curate our data based on specific experimental techniques and protocols. For instance, the same experiment conducted in different laboratories might use different measurements, such as EC50 and IC50, making it unreasonable to merge data without careful consideration of these experimental details.
> >
> > The nine datasets included in WelQrate correspond to nine therapeutic targets that were carefully chosen by domain experts. These datasets were selected to cover a range of important disease-related target classes, to ensure consistent lab experiment details, and to confirm that the experiments were conducted in a manner deemed reasonable by the domain experts. This careful selection and curation process is critical to maintaining the high quality and reliability of the benchmark datasets we provide.
> >
> > We appreciate the encouragement to expand our dataset offerings. Indeed, we are exploring ways to further automate our pipeline to extract significantly more datasets from PubChem, with the goal of leveraging these datasets to advance research on foundational molecular and drug discovery models. However, it's important to note that our aim of this work is not to build/host a large dataset collection platform like TDC, but rather to establish a benchmark standard. Also, having too many datasets could make it difficult for researchers to decide which datasets to use, thereby undermining the goal of creating a common benchmarking ground.
> >
> > If researchers were to create datasets for virtual screening in the future, our curation pipeline could serve as an inspirational model, offering a structured approach that ensures high-quality, reliable datasets. However, replicating this process with TDC datasets would be challenging due to the lack of similar experimental detail and the different nature of the tasks involved. As a result, applying our curation pipeline to TDC datasets would likely not yield the same level of data quality and consistency that we achieve with WelQrate coming from PubChem.
> >
> >
> > .

---

> > ### Author Response · Authors · 2024-08-17
> >
> > **7. In Figure 4, the authors present the results of the 2D-graph molecular representation using three different models: Graph Convolutional Neural Network (GCN), Graph Isomorphism Network (GIN), and Graph Attention Network (GAT). How do the overall performance metrics (logAUC/BEDROC) reflect the individual performance of these three models? Did the authors use mean values or another method？**
> >
> > Thank you for your question. In Figure 4, we used mean values to represent the category performance along with standard deviation. Specifically, we calculated the combined performance across all five runs for each model within that category. For example, for 2D-graph molecule representation in WelQrate we had three models (GCN, GIN, and GAT) that each had five performance values (across the five-fold cross validation), so the values reported for that bar in Figure 4 were the mean and standard deviation across 15 values.
> >
> > Note that the individual performance metrics for these GNNs are shown in Figure 6, which addresses RQ4 by comparing their performance under both random and scaffold splits. Figure 4, on the other hand, is intended to illustrate category-based performance under the random split. To avoid redundancy, we present the averaged category performance in Figure 4 rather than repeating the individual model metrics.
> >
> > We will clarify these in the final revision.
> >
> > ---
> >
> > **8. The statement '2D and 3D Graph-Based models trained on the control dataset actually outperform in logAUC and BEDROC' is somewhat fuzzy. The control dataset contains some noisy data, such as 'invalid chemical structures, inconsistent chemical representations, undefined stereochemistry, and poorly defined endpoints'. How can noisy data lead to better performance? The performance comparison seems inappropriate. It would be better to test all 9 datasets to draw stronger conclusions.**
> >
> > Our high-quality dataset filters out noise, resulting in fewer labeled data points. The noisy dataset, with more data, may act similarly to data augmentation techniques (by increasing the number of minority active samples in the dataset), where increased data volume—even with noise—can sometimes enhance training. However, despite potential benefits from noisy data in training, a rigorously curated dataset remains essential as a validation and testing benchmark to ensure model selection and that models are useful in real-world applications. In the future, we plan to investigate different methods/techniques to better train these models in the noisy/limited-labeled/imbalance data settings.
> >
> >
> > Additionally, we appreciate your suggestion to compare across all 9 datasets and have conducted this analysis. Specifically, we have performed the comparison between control dataset and WelQrate across all 9 datasets and will include all the results in the supplementary. Here we would like to highlight that out of the 20 control versus WelQrate comparisons in Figure 4 (i.e., 5 method categories (Naive, Sequence, 2D Graph, 3D Graph, Domain) and 4 metrics (LogAUC, BEDROC, EF, DCG)) we observe only 2 of the 20 have changed from what was observed in the single AID1798 dataset. Previously, we observed in AID1798 that the control was better for 2D Graph ML models in logAUC and BEDROC, but when averaged across all 9 datasets this is no longer the case (i.e., across 9 datasets 2D Graph ML models have consistently better performance when trained on WelQrate). However, the observation that logAUC and BEDROC was better for the 3D Graph ML models when trained on control compared to WelQrate still are observed across the 9 datasets, which we believe could be due to the points discussed above. Thus, in summary, for Figure 4, when converting to averaging across all 9 datasets as compared to just AID1798 we observe 18 of the 20 comparisons showing the same result.

---

> > ### Author Response · Authors · 2024-08-17
> >
> > **9. The authors hypothesize that 'this could be related to an overfitting issue in these complex models'. Why do the authors assume overfitting? Reliable ML practitioners should always check the performance difference between training and testing data to appeal robust conclusions.**
> >
> > Thank you for raising this important point. This sentense appears in the context of scaffold split, where there is a data distribution shift between training and testing. And hence we assume overfitting.
> >
> >
> > Our primary focus in this work is on the dataset and evaluation framework, with the benchmark results serving to showcase the performance of selected models on our datasets. Thus, here we conducted a standard process of model selection and hyperparameter tuning including some basic checks to understand the behavior of the models
> >
> > Specifically, we monitored the loss during training and compared it with the validation performance. As shown in Figures 3 of the attached PDF, we observed that while the training loss continued to decrease, the validation performance plateaued/fluctuated/decrease and did not improve. This behavior is suggestive of overfitting, where the model is learning patterns in the training data that do not generalize well to unseen data.
> >
> > ---
> >
> > **10. The authors mention that 'predicting the activity of molecules with unseen scaffolds represents a distribution-shift problem'. Did the authors conduct any control experiments to assess distribution shifts? Have they quantified this shift?**
> >
> > Thank you for your question. Scaffold refers to the core structure of a molecule, and it's well-known in the field that identifying active molecules with significantly different core structures from those in the training set—a process known as scaffold hopping—is challenging. In the context of our work, the scaffold split was designed specifically to test the models' ability to handle this distribution shift, where training and testing sets have very different core structures.
> >
> > To quantify this distribution shift, we conducted a analysis on the AID1798 dataset. We found that in a random split,  57.32% of the scaffolds in the test set were also present in the training set. In contrast, the scaffold split was intentionally designed so that no test molecules shared the same scaffold as those in the training set, representing a more pronounced distribution shift.
> >
> > Our control experiments to assess it are conducted in RQ4. As expected, the performance of all models dropped significantly under the scaffold split, evidenced in Figure 6 of the main paper. This performance decline highlights the challenges posed by structurally distant molecules and underscores the need for further research to improve prediction capabilities in such scenarios.

---

> ### Author Response · Authors · 2024-08-26
>
> Dear reviewer yZji,
>
> Thank you for your insightful and detailed comments.
>
> We wanted to gently remind you that the discussion period will be closing soon and kindly invite you you to review our response at your earliest convenience. We are eager to hear your feedback.
>
> Thank you once again for your time.
>
> Best regards,
>
> The Authors

---

> ### Author Response · Authors · 2024-08-29
>
> Dear Reviewer yZji,
>
> As the discussion period is coming to an end in two days, we wanted to gently remind you to review our response if you haven't had the chance yet. We greatly value your feedback and look forward to any further thoughts you may have.
>
> Thank you again for your time and consideration.
>
> Best regards,
> The Authors

---

### Official Review · Reviewer_JFZX · 2024-07-21

**Rating:** 7
**Confidence:** 4
**Correctness:** Yes
**Clarity:** Yes

**Review:**

**Quality**:
The quality of the work is high, demonstrated by a meticulous data curation process and a well-structured evaluation framework. However, the reliance on manual inspection in some parts of the curation process could introduce subjectivity, potentially affecting consistency.

**Clarity**:
The paper is clearly written and well-organized, with distinct sections that effectively communicate the methodologies and contributions. Figures and tables are used appropriately to illustrate key points and results, enhancing the reader's understanding.

**Originality**:
The introduction of WelQrate as a new gold standard for benchmarking small molecule drug discovery is an original and significant contribution. The hierarchical data curation pipeline and the emphasis on realistic evaluation metrics set this work apart from existing efforts.

**Significance**:
This work has the potential to make a substantial impact on the field of AI-driven drug discovery by providing a robust and standardized foundation for benchmarking. The public availability of the datasets, evaluation framework, and associated resources encourages transparency, reproducibility, and broader community adoption.

**Strengths:**

1. Establishes a new gold standard for benchmarking in small molecule drug discovery, enhancing the reliability and reproducibility of AI models.


2. Facilitates accurate and meaningful comparisons across different models and methodologies, promoting collaborative progress in AI and machine learning.


3. Demonstrates rigorous data curation and evaluation processes, ensuring high-quality datasets and a robust evaluation framework.

**Additional Feedback:**

No

**Documentation:**

Yes

**Limitations:**

The authors addressed the limiation

**Opportunities For Improvement:**

1. The reliance on manual inspection during data curation introduces subjectivity and potential inconsistency. Developing automated or semi-automated tools for data curation could reduce human error and improve reproducibility. For instance, implementing rule-based or machine learning algorithms to automate the detection of duplicates and application of PAINS filters could standardize the process.

2. The datasets reflect real-world scenarios with highly imbalanced classes. While this is realistic, the paper could discuss strategies for handling this imbalance in model training and evaluation. Techniques such as oversampling, undersampling, or synthetic data generation could be explored to mitigate class imbalance issues.

3. Conducting an error analysis to identify common sources of prediction errors or misclassifications would be valuable. This could help in refining both the curation process and the evaluation framework, leading to more robust and reliable benchmarks.

**Relation To Prior Work:**

Yes

**Summary And Contributions:**

The paper "WelQrate: Defining the Gold Standard in Small Molecule Drug Discovery Benchmarking" presents a new gold standard for benchmarking in small molecule drug discovery. The primary contributions are:

1. Development of a meticulously curated collection of nine datasets across five therapeutic target classes, incorporating confirmatory and counter screens, and rigorous preprocessing like PAINS filtering for high-quality data.
2, Proposal of a standardized model evaluation framework that includes featurization, 3D structure generation, and evaluation metrics for reliable benchmarking.
3. Benchmarking of various deep learning architectures (e.g., 2D/3D graph neural networks) on the curated datasets, emphasizing the impact of high-quality data curation on model performance.

---

> ### Author Rebuttal · Authors · 2024-08-17
>
> Thank you for your insightful feedback. We appreciate the opportunity to address these important points and improve our work.
>
> **1. Manual inspection during data curation may introduce subjectivity and inconsistency. Could automated or semi-automated tools, like rule-based or machine learning algorithms for detecting duplicates and applying PAINS filters, improve reproducibility and reduce human error?**
>
> Thank you for the observation. We'd like to clarify that our curation pipeline is indeed automated in applying all the necessary filters, such as the detection of duplicates and the application of PAINS filters. All automated curation Python scripts are also available on our website at WelQrate.org. The term "manual inspection" refers specifically to our process of verifying and addressing any anomalies flagged by the automated system, rather than introducing subjectivity or inconsistency.
>
> For instance, when duplicates are detected, such as two different PubChem CIDs (CID 130564 and CID 5311083) pointing to the same molecule, we don't simply discard one. Theoretically, PubChem should assign a unique CID for the same molecule. In this case, instead of automatically discarding the flagged duplicate, we manually verify the entries by reviewing the original database on PubChem’s website. Through this manual inspection, we confirmed it was indeed a mistake on PubChem’s side, where two different CIDs were incorrectly assigned to the same molecule.
>
> This manual verification step is critical to avoid potential data loss and to ensure the highest quality of the curated dataset. We will ensure that the final version of the paper makes this process clearer to prevent any misunderstandings regarding the role of manual inspection in our curation pipeline.
>
> We also thank the reviewer for suggesting automation, which has inspired the idea of developing a molecular foundational model that could further enhance the efficiency and accuracy of our curation process in the future.
>
> We appreciate the opportunity to clarify this point and will update the manuscript accordingly.
>
> ---
> **2. The datasets are highly imbalanced, reflecting real-world scenarios. Could you discuss strategies like oversampling, undersampling, or synthetic data generation to address this imbalance during model training and evaluation?**
>
> In our current work, we addressed class imbalance by employing a straightforward oversampling approach. Specifically, during model training, we used a strategy where molecules were sampled based on a probability inversely proportional to their class label frequency. This ensured that in each training batch, there was a roughly equal number of molecules from both classes, thereby mitigating the imbalance.
>
> We recognize that class imbalance is a complex problem, and while oversampling is a practical approach, it is not the only solution. In fact, our previous research has explored dedicated methods for handling imbalance in graph-based models, including imbalanced graph classification. We agree that this is an interesting area for further exploration, and we will include a discussion in the paper about potential future directions beyond oversampling, such as the use of advanced techniques like undersampling, synthetic data generation, and specialized algorithms for imbalanced data.
>
> We will also clarify in the manuscript that oversampling was the strategy applied across all models in our experiments.
>
> ---
>
> **3. Would conducting an error analysis to identify common prediction errors help refine the curation process and evaluation framework, leading to more robust benchmarks?**
>
> A detailed error analysis is an excellent suggestion and is planned for this work, but it is beyond the scope of the present manuscript, which focuses on presenting a common ground for benchmarking, including the dataset, evaluation framework and performance comparison.
>
> From the perspective of the molecules, one hypothesis is that GNNs tend to assign the same label to topologically similar molecules. However, in medicinal chemistry, even structurally similar molecules can exhibit vastly different activities, a phenomenon known as an "activity cliff." This could be a significant source of misclassification.
>
> Additionally, for molecules with structures that differ significantly from those in the training set, the model may struggle to accurately predict outcomes for this out-of-distribution testing set. This limitation highlights the challenges of generalizing to novel chemical spaces that were not well-represented in the training data.
>
> We plan to conduct a more thorough error analysis in the future to refine both the curation process and the evaluation framework, which we believe will lead to more robust and reliable benchmarks.

---

> > ### Comment · Reviewer_JFZX · 2024-08-27
> >
> > Thanks authors for the rebuttal, my concerns was addressed, will raise my score.

---

> > > ### Author Response · Authors · 2024-08-29
> > >
> > > Thank you for your positive feedback and for considering our rebuttal. We're glad we could address your concerns, and we appreciate your support.

---

> ### Author Response · Authors · 2024-08-26
>
> Dear reviewer JFZX,
>
> We greatly value your insightful and comprehensive feedback.
>
> As the discussion period is nearing its end, we kindly remind you to review our response. We are eager to receive your thoughts.
>
> Thank you again for your time and attention.
>
> Warm regards,
>
> The Authors

---

### Official Review · Reviewer_scA8 · 2024-07-25
**More clarity is required for acceptance.**

**Rating:** 5
**Confidence:** 2

**Review:**

This paper provides an evaluation framework, called WelQrate, for the performances of supervised Machine-learning models for molecular property prediction. Moreover, this work introduces nine curated datasets related to the task of molecular property prediction. Previous work is mentioned, and the contributions of the paper are clearly explained.  Code and related documentation can be accessed via the Link mentioned in the paper. The submission has some flows but does not contain any major inconsistencies. In particular, the paper requires some improvements in terms of clarity. Follows a list of the pros and cons of the paper

pros
- The paper presents a fine evaluation framework for predicting molecular properties and raises important research questions for model evaluation in broader applications.

cons.
- The paper presents issues related to clarity: For instance, no formula is provided for the evaluation metrics. No explicit explanation is given about the error bars in the reported plots. The authors mention that their framework can also be used for regression but do not mention what evaluation metrics should be considered for the regression task.
-  Data label curation procedures for regression tasks are not sufficiently motivated.
-  The experimental design presents some flows. In particular, the authors do not consider the fact that different training  and evaluation sets may lead to different performance of the same ML model, even when the test set is fixed.

**Strengths:**

**S1** The paper proposed a fine evaluation framework for methodologies addressing the task of molecular property prediction.
Moreover, the paper poses research questions (RQ1-3) that are key for model evaluation also in applications that go beyond those considered in the proposed work.

**S2** Moreover, this work introduces nine curated datasets related to the task of molecular property prediction. Importantly the datasets reflect real-world drug discovery scenarios with a low percentage of active compounds.

**Additional Feedback:**

N/A

**Clarity:**

The paper requires improvements in terms of clarity. For instance, the meaning of error bars is not explained, and the formula for evaluation metrics is not provided,

**Correctness:**

The submission has some flows, e.g. C4, but does not contain any major inconsistencies.

**Documentation:**

Code and documentation are provided and can be accessed via the Link in the paper

**Ethics:**

This work does not present any ethics-related concerns

**Limitations:**

The authors do mention  limitations of their work and encourage further research to address them.

**Opportunities For Improvement:**

**C1**  In Section 4.2  no formula is provided for the evaluation metrics. For clarity, each evaluation metric should be explicitly defined. I understand the authors want only consider evaluation metrics that can measure the ability to correctly identify true positive. However, classical classification metrics such as recall, precision and F1 would also provide meaningful insights related to the amount of true positive identified by the prediction model. The authors should consider also these classical evaluation metrics or explain in detail why they do not serve any purpose in this context.

 **C2**  The authors report error bars in Fig. 4, 5 and 6. However, no explicit explanation is given about what the error bars in each plot represent. For clarity, such information should be provided for each figure individually.

 **C3**  The authors state "WelQrate datasets also support regression tasks besides binary classification". For clarity, it should be mentioned what evaluation metrics should be considered for the regression task. Moreover, in the paper it is also mentioned that "However, not all molecules have floating-point activity values due to the cost and complexity of obtaining them experimentally and this is typically not done for inactives. So, for inactives, we set a high value (i.e., 1000) to indicate inactivity...". For clarity, it should be explained how setting such artificial values of 1000 can affect the evaluation procedures of interest for regression tasks in this application context. For instance how setting such values to 1000 would affect the interpretation of the mean absolute error, assuming that is a relevant metric?. Moreover, for clarity it should be explained how the value of 1000 was chosen and how it relates with the value range associated with those data points for which the label is known.

 **C4**  In Section 5.2, the authors state"To ensure a fair comparison, we maintained identical test sets between the control and WelQrate datasets and only vary the training and validation sets used to train the models and tune the hyperparameters." In general different training sets may lead to different performance of the same ML model. Therefore, even if the test set is fixed performance difference may be due to variation in the training and validation sets. The authors should address this issue or explain why it is not the case.

**Relation To Prior Work:**

Previous work is mentioned and the contributions of the paper are clearly explained

**Summary And Contributions:**

This paper provides an evaluation framework, called WelQrate, for the performances of supervised Machine-learning models for molecular property prediction. The framework consists of various featurization procedures, 3D structure generation approaches and evaluation metrics. Moreover, this work introduces nine curated datasets related to the task of molecular property prediction. Additionally, the authors use the proposed framework to benchmark the performances of established machine-learning approaches in the fields of drug discovery and molecular property prediction.

---

> ### Author Rebuttal · Authors · 2024-08-17
>
> Thank you for your valuable feedback. We appreciate the opportunity to address your concerns.
>
> **1. Section 4.2 lacks formulas for evaluation metrics. Classical metrics like recall, precision, and F1 could provide additional insights**
>
> Due to space constraints in the main paper, we included the formulas for metrics $log𝐴𝑈𝐶[0.001,0.1]$, $EF_{100}$, and $DCG_{100}$ in the supplement. We acknowledge that we missed providing the formula for $BEDROC$, which is included here:
>
> $BEDROC$, bounded by the interval [0,1], emphasizes the model's ability to rank active compounds early in the prediction list. It is derived from the robust initial enhancement ($RIE$), its minimum value $RIE_{min}$ (when all the actives are ranked at the tail of the list), and its maximum value $RIE_{max}$   (when all the actives are ranked at the beginning of the list), defined as follows:
>
> $$RIE = \frac{\frac{1}{n}\sum_{i = 1}^ne^{- \alpha x_i}}{\frac{1}{N}\left(\frac{1 - e^{- \alpha}}{e^{\alpha / N} - 1}\right)}$$
> $$RIE_{max} = \frac{1 - e^{- \alpha R_a}}{R_a\left(1 - e^{- \alpha}\right)}$$
> $$RIE_{min} = \frac{1 - e^{\alpha R_a}}{R_a\left(1 - e^{\alpha}\right)}$$
>
> Where *n* and *N* are the numbers of actives and total compounds tested, respectively. $x_i$ is the relative rank of the $i$th active such that $x_i=r_i/N$ for $r_i$ being its rank in the prediction list. $R_\alpha$ is the ratio of actives $(n/N)$, and $\alpha$ is a tunable parameter that controls the metric's sensitivity to early recognition. we used the recommended value of $\alpha$=20 as suggested in the original paper. The $BEDROC$ score is calculated as:$$BEDROC = \frac{RIE - RIE_{min}}{RIE_{max} - RIE_{min}}
> = \frac{\sum_{i = 1}^n - e^{r_i/ N}}{\frac{n}{N}\left(\frac{1 - e^{- \alpha}}{e^{\alpha / N} - 1}\right)} \times \frac{R_a\sinh \left(\alpha / 2\right)}{\cosh \left(\alpha / 2\right) - \cosh \left(\alpha / 2 - \alpha R_a\right)} + \frac{1}{1 - e^{\alpha (1 - R_a)}}
> $$
>
> Regarding the use of classical metrics like recall, precision, and F1-score, we want to emphasize that our metrics are tailored to reflect the practical needs of real-world drug discovery. In such contexts, the cost of synthesizing or purchasing molecules for further testing is significant, e.g., 100 molecules would cost around \$50,000. Due to budget/labor constraints, researchers can only afford to test a limited number of top-ranked molecules from the prediction list, so the main objective is to recommend to them a top ranked set of molecules for further investigation/testing.  Actually, this scenario is akin to search result retrieval/recommender systems, where the evaluation metrics only consider the performance in the top-ranked set, since users would not consider the quailty of predictions in the lower rankings just as the domain experts using these tools here would only typically care about the performance in the top-ranked set (that they would be intending to purchase for further testing).
>
> Thus, our focus is not solely on identifying all true positives but on ensuring that the true positives are ranked high in the prediction list, which is crucial for efficient resource allocation in drug discovery. For example, this is why metrics like $BEDROC$ (that prioritize early recognition of actives) and $DCG_{100}$ (that only considers the top 100 scored/ranked molecules in the testing data for evaluation) are particularly relevant over metrics that consider the performance across all testing set samples.
>
> ---
>
> **2. Error bars in Figures 4, 5, and 6 are not explained.**
>
> The error bars represent standard error across multiple experimental runs, accounting for variability. This clarification will be included in the final version of the paper.

---

> > ### Author Response · Authors · 2024-08-17
> >
> > **3.  Please specify appropriate evaluation metrics for regression task. Additionally, clarify how setting an artificial value of 1000 for inactives impacts regression evaluation and how it was chosen.**
> >
> > Our data is curated from actual biochemical experiments stored and detailed on PubChem database. The floating-valued labels correspond to measured activity values, typically expressed as EC50 or IC50 in the unit of micromolar (µM). These measurements, requiring extensive biochemical experiments, are more expensive to obtain compared to binary active/inactive labels, so they are usually only available for confirmed active compounds to manage costs effectively. For other compounds, the data remains binary (active/inactive).
> >
> > In our three regression datasets, the active molecules generally have activity values in the tens of micromolar range (details are in the README file at the website data download link). To represent inactive compounds in these datasets, we chose to set a value of 1000 µM (or 1 mM), as it is commonly considered inactive in drug discovery contexts.
> >
> > We acknowledge that setting this artificial value of 1000 µM could skew the results. The chosen value of 1000 µM was selected as a default to indicate inactivity, but it may not be optimal for all use cases.
> >
> > We suggest that researchers consider using weighted Mean Squared Error (weighted MSE) for regression tasks. In weighted MSE, more weight is applied to data points with high activities and less to those with lower ones.
> >
> > We encourage researchers with a deep understanding of drug discovery to leverage the floating-value information in the dataset and consider adjusting the default value of 1000 µM according to their specific needs. For instance, they might choose to set this value to the maximum detectable activity or another relevant threshold. For other researchers, as noted in the paper, we recommend sticking to binary classification tasks, especially if they are not comfortable with the implications of using this artificial value.
> >
> > We will ensure that the final version of the paper clarifies these points to guide researchers in selecting the most appropriate evaluation approach for their work.
> >
> > ---
> >
> > **4. Section 5.2 mentions identical test sets but different training/validation sets. Could performance differences arise from this?**
> >
> > The paper's focus is on the impact of high-quality datasets. By training on datasets of varying quality and testing on a fixed high-quality set, we assess how models derived from different data qualities approximate real-world performance. The fixed high-quality test set serves as the closest reality check, highlighting the value of better training data.

---

> > > ### Comment · Reviewer_scA8 · 2024-08-26
> > > **Thank you for your rebuttal**
> > >
> > > I have further questions and comments related to the evaluation metrics for regression.
> > >
> > > Using Weighted MSE may be a valid solution. However, the methodology provided to set the weights is only qualitative: higher weights for data points with higher activities. No quantitative standard is given to determine the weight value. Moreover,  even assuming a given quantitative procedure for setting the weights, why is weighted MSE the only mentioned option? For instance, would weighted mean absolute error not be an equally valid option?
> > >
> > > I think that evaluation procedures for regression should be investigated more. If WellQrate is to be considered a new gold standard for drug discovery and benchmarking, the proposed evaluation procedures should be thoroughly investigated, and their implementations should be clearly defined, carefully interpreted and explained. I do not think that is the case for regression.

---

> > ### Author Response · Authors · 2024-08-29
> > **Thank you for your insightful suggestions, which have inspired us to explore a new direction**
> >
> > Thank you for your further questions and comments regarding the evaluation metrics for regression.
> >
> > We reviewed existing literature on regression tasks in molecular activity prediction and found relevant studies, though they do not specifically address the challenges we face in this context, such as the need for accurate top-ranked true actives and handling missing values in inactives. These challenges require particular attention because they are crucial for the effectiveness of molecular activity prediction models. Existing approaches often focus on general predictive accuracy, but may not fully capture the nuances required for our specific application.
> >
> > As you suggested, there are multiple alternatives for addressing this scenario. Drawing inspiration from metrics used in search retrieval, where top-ranked predictions are prioritized, and from imbalanced datasets, where different weights are assigned to predictions, we propose to study the following metrics (with the potential for more). These metrics broadly fall into two categories: value-based metrics, which emphasize accurately predicting the values of top-ranked true actives, and rank-based metrics, which focus on the correct ordering of predictions rather than their exact values (considering that experimental errors might render the floating-number values less precise).
> >
> > In the following fomulas, $y_i$ denotes the $i$th ground truth and $\hat{y_i}$ denotes the corresponding prediction
> >
> > ---
> >
> > ### 1. Value-Based Metrics:
> >
> > ---
> >
> > **Weighted Mean Squared Error (WMSE)**
> >
> > Gives higher weights to true active predictions that are ranked higher.
> >
> >
> > $\text{WMSE}$ =  $\frac{\sum{w_i}\cdot(y_i-\hat{y}_i)^2}{\sum{w_i}}$
> >
> > (* the index $i$ and upper limit $n$ of the summation in the equation are omitted due to the formula display error on OpenReview. Same for WMAE.)
> >
> >
> > **Weighted Mean Absolute Error (WMAE)**
> >
> > Similar to WMSE but focuses on absolute differences, potentially reducing the impact of outliers.
> >
> > $\text{WMAE} = \frac{\sum w_i \cdot |y_i - \hat{y}_i|}{\sum w_i}$
> >
> > ---
> >
> > ### 2. Rank-Based Metrics:**
> >
> > ---
> >
> > **Regression $DCG_{100}$ ($rDCG_{100}$)**
> >
> > This metric is similar to the binary $DCG_{100}$ defined in the supplement, but instead of a binary relevance score, the score is based on the proximity of the prediction to the ground truth: when the prediction is close to the ground truth, the relevance score is high; otherwise, it approaches zero..
> >
> > To calculate DCG, a simpler version metric named Cumulative Gain (CG) [1] is introduced below. CG is the sum of the relevance value of a compound in the selection set
> >
> > $CG_{100}=\sum_{i=1}^{100}rel_i$
> >
> > where the $rel_i$ is the relevance score, and is defined as:
> >
> > $rel_i$ = $sigmoid(-log|{y_i-\hat{y}}|)$
> >
> >
> > It can be observed that $CG_{100}$ is unaffected by changes in the ordering of compounds. $rDCG_{100}$ hence aims to penalize a true active molecule appearing lower in the selection set by logarithmically reducing the relevance value proportional to the predicted rank of the compound, i.e.,
> >
> > $rDCG_{100} = \sum_{i=1}^{100}rel_i/log_{2}(i+1)$
> >
> >
> >
> > **Spearman's Rank Correlation (SRC)**
> >
> > Measures the correlation between the predicted rankings and the true rankings, emphasizing the correctness of the order rather than the specific values.
> >
> >
> > $SRC=1-\frac{6\sum(d^2_i)}{n(n^2-1)}$
> >
> > where $d$ = difference between the two ranks of each observation
> > $n$ = number of observations
> >
> > ---
> >
> > We acknowledge that the development of a standardized set of evaluation metrics for regression tasks in molecular activity prediction requires careful consideration and further investigation. This work aims to initiate a discussion on the complexities involved in these tasks, and we recognize that additional research is needed to refine the standards.
> >
> > In the current study, we will include a dedicated discussion section to highlight the challenges posed by regression tasks, particularly in terms of accurate top-ranked predictions and the presence of missing values in inactives. We also plan to explore these metrics further in future work and refine our approach as we continue to develop WelQrate as a gold standard for drug discovery and benchmarking.
> >
> > Reference:
> >
> > [1] Järvelin, Kalervo, and Jaana Kekäläinen. "IR evaluation methods for retrieving highly relevant documents." ACM SIGIR Forum. Vol. 51. No. 2. New York, NY, USA: ACM, 2017.

---

> ### Author Response · Authors · 2024-08-26
>
> Dear reviewer scA8,
>
> Thank you for your insightful and detailed review comments!
>
> We want to raise a kind reminder that the discussion period will be closing soon, and we kindly invite you to review our response. We are looking forward to your feedback!
>
> Thank you again for your time.
>
> Best,
>
> Authors

---

### Decision · Program_Chairs · 2024-09-26

**Decision:**

Accept (Poster)

**Comment:**

This paper proposes a new gold standard for small molecule drug discovery benchmarking, WelQrate, which consists of three components: 1) Data Curation Pipeline, with which a collection of 9 datasets spanning 5 therapeutic target classes is curated. 2) Evaluation Framework. 3) Benchmarking: existing representative deep learning architectures (e.g., 2D/3D graph neural networks) on WelQrate are evaluated. The reviews are a little mixed, with the scores being 5, 7, 4, and 8. The major concern is the so-called “gold-standard.” It seems there is no very convincing evidence for this claim. Another concern is that we are now in a deep learning and LLM age, it is not quite sure about the value and utility of such a benchmarking.